

# Winter stratification phenomena and its consequences in the Gulf of Finland, Baltic Sea

Taavi Liblik, Germo Väli, Inga Lips, Madis-Jaak Lilover, Villu Kikas, Jaan Laanemets

Department of Marine systems at Tallinn University of Technology, Estonia

*Correspondence to*: Taavi Liblik (Taavi.liblik@taltech.ee)

**Abstract.** Stratification plays an essential role in the marine system. The shallow mixed layer is one of the

preconditions for the enhanced primary production in the ocean. The general understanding is that the mixed layer is well deeper than the euphotic zone in the Baltic Sea during winter. In this work, we demonstrate the wintertime stratification is a common phenomenon in the Gulf of Finland. Shallow haline stratification at the depth comparable to the euphotic zone depth forms in late January–early February. Stratification is evoked by the positive buoyancy flux created by the westward advection of riverine water along the northern coast of the gulf. Fresher

water and haline stratification appeared approximately one month later in the southern part of the gulf. The phenomena can occur in the whole gulf and also without ice. Chl *a* concentration and phytoplankton biomass in winter can be comparable to mid-summer. The limiting factor for phytoplankton bloom in winter is likely insufficient solar radiation.

## 1.   Introduction

The Baltic Sea is a shallow, brackish sea with limited water exchange with the North Sea. The sea has strong seasonality and gradients of oceanographic parameters (Leppäranta and Myrberg, 2009). Upper mixed layer with a typical depth of 10–20 m forms in spring and is separated from the rest of the water column by seasonal

thermocline. The mixed layer warms up to 15–24 °C (Stramska and Białogrodzka, 2015; Tronin, 2017) and thermal stratification strengthens until August. The thermocline is eroded by thermal convection, wind stirring, and current shear, and the mixed layer deepens down to the sea bottom or the halocline at 40–80 m depth in autumn-winter (Lass et al., 2003; Liblik and Lips, 2017; Väli et al., 2013). This annual cycle in stratification has substantial implications on physical, biogeochemical and biological processes in the sea. First, characteristics of the

pycnocline (e.g. strength) determine vertical fluxes between surface and sub-thermocline layer. Moreover, vertical structure of currents is strongly linked to pycnoclines (Suhhova et al., 2018). The annual cycle in stratification, together with solar radiation, mainly determines the seasonality in primary production and nutrient consumption. Vertical mixing from the deeper layers and low production in winter allows the accumulation of nutrients in the upper layer (e.g. Lilover and Stips, 2008; Nehring and Matthäus, 1991). In spring, when the water column becomes

stable, mixed layer depth is shallower than euphotic zone, and solar radiation is strong enough, spring bloom is evoked (Fleming and Kaitala, 2006; Jaanus et al., 2006; Lips et al., 2014; Wasmund et al., 1998).

Stratification in the northeastern Baltic Sea is particularly strong and variable. The largest river in the





Baltic Sea catchment area, Neva, discharges to the eastern end of the Gulf of Finland. Mean river runoff to the gulf is 3700 $m^3 s^{-1}$ (Johansson, 2018). Since the river discharge is concentrated to the east and the gulf is connected to the Baltic Proper in the west, mean longitudinal salinity gradient from virtually 0 g $kg^{-1}$ in the easternmost end to 6 g $kg^{-1}$ in the west exists in the upper layer of the gulf (Alenius et al., 1998). Due to mean cyclonic circulation in the upper layer and prevailing westward current along the northern coast (Palmen, 1930; Rasmus et al., 2015; Stipa, 2004) mean salinity in the upper layer is lower along the northern coast of the gulf compared to the southern coast (e.g. Kikas and Lips, 2016). Gulf has a free water exchange with the Baltic Proper. Thus, quasi-permanent halocline and saltier deep layer also exist in the gulf. This lateral and vertical structure can be strongly modified by wind forcing. Westerly winds accumulate saltier upper layer water, deepen the upper mixed layer depth in the gulf (Liblik and Lips, 2017) and cause weakening of the halocline (Elken et al., 2003). This process can lead to the complete mixing of the water column in the gulf (Elken et al., 2014; Liblik et al., 2013; Lips et al., 2017). Easterly winds, in turn, encourage westward transport of riverine water and strengthen haline stratification in the whole water column (Liblik and Lips, 2017). Wind-driven processes also generate considerable across-gulf inclination of the pycnoclines (Liblik and Lips, 2017), and upwelling and downwelling events along the southern and northern coast (Kikas and Lips, 2016; Lehmann et al., 2012; Lips et al., 2009).

The northeastern part of the Baltic Sea is ice-covered every winter (e.g. Uotila et al., 2015), the extent of the ice-coverage has high inter-annual variability. The brackish water of the Baltic Sea has maximum density temperature $T_{md}$ (2.2–3.3 °C) higher than freezing temperature (from –0.4 to –0.1 °C), unlike in most of the world ocean. Thus, warming of the surface layer below $T_{md}$ increases water density and causes convection and vertical mixing. Cooling at $< T_{md}$ stabilizes the water column. The temperature typically passes the $T_{md}$ in the northern and eastern part of the Baltic Sea during the cooling period (Karlson et al., 2016; Liblik et al., 2013) while it is not always the case in the offshore areas in the southern Baltic Sea (e.g. Stepanova et al., 2015). Lateral haline buoyancy flux can compensate the thermal convection and stabilization of the shallow upper layer in spring can occur at temperatures already below $T_{md}$ (Eilola, 1997; Eilola and Stigebrandt, 1998; Stipa et al., 1999). One reason for the latter is the relatively low thermal expansion at temperatures around $T_{md}$, i.e. temperature impact on density is relatively small compared to salinity. Thus, the onset of the seasonal pycnocline is not necessarily initiated by the thermal buoyancy but could be related to the haline buoyancy instead. Straightforward evidence of the latter is the temperature below $T_{md}$ in the cold intermediate layer after the establishment of the seasonal pycnocline (Chubarenko et al., 2017; Eilola, 1997; Liblik and Lips, 2017). Without haline stratification, warming would cause mixing until $T_{md}$ is reached. The haline stratification creates favourable conditions for spring phytoplankton bloom (Kahru and Nõmmann, 1990; Lips et al., 2014). Haline stratification under ice has been observed, in the vicinity of River Siikajoki mouth in the Bothnian Bay (Granskog et al., 2005), at Tvärminne in the northwestern Gulf of Finland (Merkouriadi and Leppäranta, 2015) and the Himmerfjärden bay in the western Baltic Proper (Kari et al., 2018). Ice coverage prevents wind mixing and therefore even relatively small river runoff can form fresher water plume and stratification, which can reach 10–20 km from the river mouth. Mentioned above studies (Granskog et al., 2005; Kari et al., 2018; Merkouriadi and Leppäranta, 2015) have dealt with winter and early spring haline stratification topics locally in nearshore regions and near relatively small freshwater sources.

Details about the formation of the haline stratification in the larger areas of the Baltic Sea during wintertime is mainly unknown. The Gulf of Finland has favourable preconditions for that process in winter. Gulf receives large amounts of freshwater, and it is at least partly covered by ice during winters. The present work


hypothesizes that haline stratification depth comparable to the euphotic zone depth occurs in the Gulf of Finland and potentially in the northeastern Baltic Proper during wintertime. It means that the general understanding that

the water column is mixed down to the halocline in the open Baltic Sea in winter (Leppäranta and Myrberg, 2009) might not be valid in the northeastern Baltic Sea. To testify this hypothesis, we analyzed the research vessel gathered and autonomously acquired data along and across the Gulf of Finland. Likewise, model simulation data were examined.

## 2. Data and methods

### 2.1. In-situ and remote sensing data

Two measurement campaigns we arranged in winters 2011/12 and 2013/14 to investigate estuarine

circulation reversals in the Gulf of Finland (Liblik et al., 2013; Lips et al., 2017). Details of the along the gulf surveys and data processing are well described in the referenced papers. Six surveys onboard the RV Salme were conducted in both winters. In this work, we present temperature and salinity data from the cruises on 21 December 2011, 24–25 January, 7–8 February, 29 February, 15–16 March 2012 and 9–10 January, 3–4 February and 4–5 March 2014. The vertical profiles of temperature, salinity, and chlorophyll a (Chl a) fluorescence were recorded

using an Ocean Seven 320plus CTD probe (Idronaut S.r.l.) equipped with a Seapoint Chl a fluorometer. The quality of the salinity data was calibrated against thewater sample analyses using a high-precision salinometer 8410A Portasal (Guildline). The mean difference and standard deviation of salinity measured by CTD and salinometer was -0.022±0.014 g kg$^{-1}$ in 2011/2012 and -0.009±0.009 in 2013/2014. Thus, after removal the offsets the accuracy of salinity data was 0.02 g kg$^{-1}$. Temperature sensors were calibrated before and after surveys in the Idronaut

factory and the differences with calibration device were smaller than initial accuracy (0.001 °C) of the Ocean Seven 320plus temperature sensor.

Chlorophyll *a* (Chl *a*) fluorescence data was compared and calibrated against water samples in the selected cruises (Fig 2). The Chl *a* concentration in the water samples was determined using Whatman GF/F glass fibre filters following extraction at room temperature in the dark with 96% ethanol for 24 h. The Chl *a* content

from the extract was measured spectrophotometrically (Thermo Helios g) in the laboratory (HELCOM, 1988). Phytoplankton biomass was determined from the water samples in winter 2014, and the samples were not collected in all stations (Fig. 1). The sub-samples (100ml) were preserved and analyzed according to the HELCOM recommendations and EVS-EN 15972:2011 standard. The phytoplankton carbon (C) content was calculated using C: biovolume factors according to Menden-Deuer and Lessard (2000) and for photosynthetic naked ciliate

Mesodinium rubrum according to Putt and Stoecker (1989).

The wind data were measured at Tallinnamadal Lighthouse (Fig. 1) at the height of 36 m above the sea level and recorded with 1-h interval. The wind speed was multiplied by a height correction coefficient of 0.91 (neutral atmospheric stratification) to reduce the recorded wind speed to that of the height of 10 m (Launiainen and Saarinen, 1984).

Tallinn–Helsinki Ferrybox measurements of temperature and salinity from January-March 2012, 2014





and 2016 are presented in the paper. The information about the Ferrybox system and data processing are given in detail in Kikas and Lips (2016). The analyses have shown that a correction of 0.08 g kg$^{-1}$ (the value has been stable over the years) must be added to the recorded salinity (Kikas and Lips, 2016). After removal the offset the standard deviation of the difference between salinity measured by Ferrybox and Portasal salinometer was 0.01 g kg$^{-1}$. The

accuracy of temperature sensor of the Ferrybox is 0.04 °C (Kikas and Lips, 2016).

Historical data collected by the Department of Marine Systems at Tallinn University of Technology and the ICES HELCOM dataset (https://ocean.ices.dk/helcom/) were used to describe the stratification conditions in the past. Quality assurance and processing of this data were in accordance with the HELCOM Monitoring Manual (Anon, 2017).

Remotely sensed OSTIA (Donlon et al., 2012) mean sea surface temperature (SST) data for the period 2010–2019 were obtained from the Copernicus marine environment monitoring service products. Mean difference of the remotely sensed SST product and in-situ measurements is 0.01-0.03 °C and standard deviation 0.4-0.5 °C (Worsfold et al., n.d.). The daily mean SST along the thalweg in the Gulf of Finland (Thalweg GoF in Fig. 1) and in the Gotland Deep (box in Fig. 1) were calculated to determine if and when the SST declined below and rose

above $T_{md}$. Salinity 6 g kg$^{-1}$ and 7 g kg$^{-1}$ were used in the $T_{md}$ estimation for the Gulf of Finland and Gotland Deep, respectively.

Density is given as a potential density anomaly ($\sigma_0$) to reference pressure of 0 dbar (Association for the Physical Sciences of the Sea, 2010). The upper mixed layer (UML) depth was defined as the minimum depth where $\rho_z \geq \rho_3 + 0.15$ kg m$^{-3}$ was satisfied. The density at 3 m depth is $\rho_3$, and $\rho_z$ is the density at a certain depth $z$.


### 2.2.    1.2. Modelling

The General Estuarine Transport Model (GETM, (Burchard and Bolding, 2002)) was applied to obtain UML parameters in the Gulf of Finland and Eastern Baltic Sea. GETM is a primitive equation 3-dimensional, a

free surface hydrostatic model with the embedded vertically adaptive coordinate scheme (Gräwe et al., 2015; Hofmeister et al., 2010)

In GETM the vertical mixing parameters are calculated using a General Ocean Turbulence Model (GOTM, (Umlauf and Burchard, 2005). In this model, the eddy diffusivity and eddy viscosity parameters are found by a two-equation k-ε model coupled with an algebraic second-moment closure (Burchard et al., 2001; Canuto et

al., 2001).

The horizontal resolution of the GETM grid is 0.5 nautical miles (approximately 926 m) over the whole Baltic Sea (Fig. 1); there are 60 adaptive layers in the vertical direction. The digital topography of the Baltic Sea was taken from Baltic Sea Bathymetry Database (http://data.bshc.pro/) and adapted for the Gulf of Finland based on the data by Andrejev et al. (2010). The atmospheric forcing (the wind stress and surface heat flux components)

was calculated from the wind, solar radiation, air temperature, total cloudiness and relative humidity data generated by HIRLAM (High-Resolution Limited Area Model) version maintained by the Estonian Weather Service with the spatial resolution of 11 km and the daily forecast interval of 1 h with total forecast length 54 h (Männik and Merilain, 2007). The wind velocity components at the 10 m level along with other HIRLAM meteorological

parameters were interpolated to the model grids. The model simulation runs were performed from 1 April 2010 to
31 December 2019.

The model domain has an open boundary in the Danish straits. For the boundary conditions, the sea surface height measurements from the Gothenburg station and the climatological temperature and salinity profiles along the open boundary were utilized. The freshwater input from 54 largest Baltic Sea rivers, together with their interannual variability reported in the HELCOM (Johansson, 2018) was used.

The initial thermohaline field was generated by COPERNICUS reanalysis of the Baltic Sea for the period 1989–2014. The product provided the horizontal resolution of 3 nautical miles and the vertical resolution from 5 m at the surface up to 50 m in the near-bottom layers.

For the model validation available Ferrybox data (2011–2016) along the transect from Tallinn to Helsinki (see Fig. 1 for location) were used. The model captures the observed variability of the temperature and salinity reasonably well (Fig. 3). Standard deviations of the simulated temperature and salinity for the overall (1.11. 2011– 1.06. 2016) and wintertime (December–March 2011–2016) periods are close to observations. Standard deviation of simulated salinity is smaller than observed in winter 2016 (January–March) and larger in 2012, while in 2014 it is close to observations. The simulated variability of the temperature is captured well – the standard deviations from the simulations are at least 0.8 of the observed for all the time-periods. The model slightly overestimated the temperature variability for the winter of 2012.

For salinity, the overall correlation coefficient is 0.62, while it is over 0.74 both for the whole wintertime period and single years as well. A higher correlation (as expected) is for temperature. Overall correlation as the seasonal variability is included, is 0.99, while for the wintertime it is 0.95. Very high correlation (>0.94) for the temperature is also for single winters.

The mean root-squared differences between model and observed values are slightly larger for the salinity but do not exceed the observed variability. In general, the model captures the wintertime changes in the surface layers of the Gulf of Finland well. More details about the model setup and validation in the Baltic Proper is given in (Zhurbas et al., 2018).

## 3. Results

### 3.1. Wind forcing, hydrography and Chlorophyll *a* patterns

Strong westerly wind, maximum along-gulf wind stress was 1.3 N m$^{-2}$, prevailed before the survey on 21 December 2011 (Fig. 4a). The cumulative wind stress increased from 1 November to 21 December by 6 N m$^{-2}$ d. As a result, warm (>5 C°, Fig. 5a), relatively salty (>6.3 g kg$^{-1}$, Fig. 5b) and well-mixed water column was observed in the gulf (Fig. 5c). Very low Chl *a* concentration, around 1 mg m$^{-3}$, was seen in the section (Fig. 5d). Weaker easterly winds prevailed since mid-January before the survey on 24–25 January 2012 (Fig. 4a). Lower temperature (3–4 C°, Fig. 5e) in the upper 20 m coincided with slightly fresher water on 24–25 January 2012 (Fig. 5f). Salinity minimum (down to 5.8 g kg$^{-1}$) caused stratification in the upper layer (Fig. 5g) at the distance of 80 to 110 km in the section, and slightly higher Chl *a* concentration (up to 1.5 mg m$^{-3}$) was seen there (Fig. 5h).





Variable and relatively weak winds prevailed in late January and early February (Fig. 4a). The temperature was lower than $T_{md}$ (2.7 °C) in the upper layer on 7–8 February (Fig. 5i). The cold water in the upper layer coincided with lower salinity (4.8–6.0 g kg$^{-1}$, Fig. 5j) and remarkable stratification was observed (Fig. 5k). Higher Chl $a$

concentration, occasionally >2 mg m$^{-3}$ was seen in the fresher and cold water along the section (Fig. 5l). Lateral Chl $a$ shape was closely linked to the salinity (density) structure. Higher Chl $a$ concentration was connected to the lower salinity and lower Chl $a$ concentration to the higher salinities. Westerly winds prevailed during the period before the next survey at the end of February (Fig. 4a). This resulted in well-mixed conditions and relatively high salinity (6.0–6.7 g kg$^{-1}$) in the western part of the section on 29 February (Fig. 5m–n). Lower salinity, stronger

stratification and slightly higher Chl $a$ in the upper layer were observed in the central part of the section (Fig. 5n–p). The Eastern part of the section was not visited due to ice conditions on 29 February. In the middle of March (15–16 March) the water temperature was still well below $T_{md}$, and strong haline stratification was observed along the whole transect (Fig. 5q–s). Concentrations of Chl $a$ within the range of 2–4 mg m$^{-3}$ were found in the upper layer (Fig. 5t).

Similar developments in the wind forcing and spatiotemporal patterns in temperature, salinity, density and Chl $a$ were observed in winter 2013/14. Strong westerly wind dominated until early January 2014, the cumulative wind stress increased since 1 November 2013 by 10 N m$^{-2}$ d (Fig. 4b). Well-mixed water column and low Chl $a$ were observed on 9–10 January 2014 (6a–d). Fresher and colder water, but only slightly higher Chl $a$ was found in the upper layer on 3–4 February (Fig. 6e–h). Spreading of fresher water (salinity <6 g kg$^{-1}$) and the

stronger stratification were observed in most of the section on 4–5 March (Fig. 6j, k). Higher Chl was found in the cold and fresher upper layer, especially in the eastern part of the section (Fig. 6il).

To illustrate wintertime re-stratification phenomena and formation of the shallow upper mixed layer along the gulf, we show longitudinal upper mixed layer depth, surface layer salinity, the density difference between 40 m depth and the surface layer, and surface layer Chl $a$ in winters 2011/12 and 2013/2014. Very thick mixed layer

(70–90 m), high surface salinity (6.3–6.5 g kg$^{-1}$), small along gulf gradient of the surface salinity, small density difference (<0.1 kg m$^{-3}$) and low Chl $a$ concentration was observed on 21 December 2011 (Fig. 7). Similar characteristics were observed in most of the section on 24 January 2012. An exception was the region at a distance from 80 to 110 km, where surface salinity within the range 5.8–6.3 g kg$^{-1}$ was observed. Interestingly, saltier water was found further in the east again. Since freshwater originates from the east, the fresher water must have been

first flown to west along the northern coast and later advected to the central part of the gulf likely as a filament. Upper mixed layer depth < 20 m, density difference up to 0.5 g kg$^{-1}$ and Chl $a$ concentration up to 1.5 mg m$^{-3}$ were observed in this area. Mixed layer depth <20 m and density difference >0.5 g kg m$^{-3}$ occurred in most of the sections on 7–8 February, 29 February and 15–16 March 2012. Similar developments were seen in winter 2013/14. Thick mixed layer, high salinity, small density difference and low Chl $a$ were observed on 9–10 January 2014.

Occasionally lower salinity, smaller upper mixed layer thickness, stronger stratification and elevated Chl $a$ concentration were found on 3–4 February, and well-developed stratification and Chl $a$ concentration mostly in the range 2–3.5 mg m$^{-3}$ registered on 4–5 March 2014.

Thus, we observed haline stratification and elevated Chl $a$ the beginning of February in both winters (2011/12 and 2013/14). Shallow mixed layer depth (<20 m) was not observed after prevailing westerly winds and

in the case when sea surface temperature was >$T_{md}$. Stratification formed as fresher water occupied the upper layer.





Next, we describe across the gulf changes of temperature and salinity using measurements acquired by the Ferrybox system at Tallinn–Helsinki transect in January–March 2012, 2014 and 2016 (Fig. 8). General temporal changes of salinity and temperature along the transect in these years were quite similar as was wind forcing (Fig. 4). According to the observations at longitudinal sections (Figs. 5 and 6), we assume sea surface salinity of 6 g kg$^{-1}$ as highest salinity, where stratification and relatively shallow upper mixed layer could form. Similarly to along-gulf observations (Fig. 5a, b), salty and warm water occupied the transect at the beginning of January in 2012 (Fig. 8a, b). Fresher water (< 6 g kg$^{-1}$) spreading from the east to west covered the northern part of the transect by the end of January while salinity slightly increased in the southern part of the section at the same time. The area covered by fresher water widened to almost the entire section by mid-February. Water temperature declined below $T_{md}$ in the northern part in the first half of January while in the central and southern part of the section temperature dropped below $T_{md}$ by the end of January. A similar spatiotemporal pattern in the sea surface salinity was observed in 2014 and 2016 (Fig. 8c–f). Fresher water first appeared in the northern part in early- or mid-January. The segment covered by fresher water widened during January and most of the transect was occupied by water with salinity < 6 g kg$^{-1}$ at the end of January 2016 and in the mid-February 2014. Strong westerly wind impulse occurred at the end of January–beginning of February (Fig. 4c). We suggest the lighter, less saline water that originates from the east flowed along the northern coast to the west and was later transported toward the southern coast in the central and western part of the gulf. Latter is likely related to the Ekman transport induced by westerly winds early February in both years (Fig. 4). Thus, stratification related to the spreading of fresher water forms about one month earlier in the northern part than in the southern part of the gulf.

### 2.3.     3.2. The occurrence of re-stratification phenomena

The spatiotemporal pattern of the re-stratification process can be described by model simulation data and statistics of historical observations. Monthly mean simulated UML depth and occurrence of the UML depth <20 m are presented in Fig. 9.

Generally, the occurrence of UML depth <20 m was very low and mean UML depth varied within the range 40–60 m in the western and central gulf in November, December and January 2010–2019 (Fig. 9a–f). As an exception, the probability of the UML depth <20 m was 30–40% in the northern part of the eastern area in January. In February, the occurrence of UML depth <20 m increased to 50–60% (Fig. 9g and h). Still, in the southern and western parts of the gulf, the mean UML depth was 30–40 m in February. These statistics based on model simulation data well agree with our observations of westward advection of fresher water from the northern coast (Fig. 8). The mean UML depth was 20 m or lower in the central part of the gulf while the UML was thicker at the entrance of the gulf in March (Fig. 9i). The occurrence of the UML depth <20 m was >60% in the central part, around 50% at the entrance of the gulf and much less in the west from the longitude 22° E (Fig. 9j). A similar pattern revealed in the mean occurrence of the density difference between 40 m and sea surface >0.5 kg m$^{-3}$, based on in-situ measurements in 1904–2020 (Fig. 10). The occurrence was 40–75% in the central part of the gulf, 30–50% in the western part of the gulf and <5% further in the west. Thus, the wintertime upper layer stratification extended to 23°E in the west.

To describe the development of UML depth from October to March along the transect from the northern


Baltic Proper to the central Gulf of Finland and in the Gotland Deep model simulation data (2010–2019) were used. The mean UML depth for the transect in the gulf (Thalweg GoF, Fig. 1) and Gotland Deep (box, Fig. 1) was calculated. The maximum of the mean UML depth in the gulf mostly occurred in December and well before SST decreased to $T_{md}$ (Fig. 11a). The onset of re-stratification occurred at temperatures below $T_{md}$ (Fig. 11a). The temperature dropped below $T_{md}$ later and raised over $T_{md}$ earlier in the Gotland Deep compared to the gulf. In five

winters out of ten SST did not fall below $T_{md}$ in the Gotland Deep (Fig. 11a). However, whether the temperature was below $T_{md}$ or not, re-stratification phenomena in the upper layer did not occur in the Gotland Deep in January– March. It means that buoyancy, created by slight thermal stratification at $<T_{md}$ is overshadowed by vertical mixing in the Gotland Deep. Vertical mixing also dominated in the Gulf of Finland in November–December. Still, from late January or early February, the advection of fresher water (Fig. 8) creates a shallow mixed layer (Fig. 11a).

Time-series of simulated UML depth along the transect from the northern Baltic Proper to the central Gulf of Finland from October to March in 2010–2019 showed considerable inter-annual variability (Fig. 11b). The deepest UML occurred in the gulf in winters 2011/12 and 2013/14, i.e. precisely the years, when measurements along the thalweg also showed deep UML in the gulf (Figs. 5 and 6). The estuarine circulation reversal caused by strong westerly winds gave rise to a deep UML while the re-stratification occurred after the prevailing of easterly

winds (Figs. 4–6). The frequency of westerly (easterly) winds over the Gulf of Finland in winter is positively (negatively) correlated to the North Atlantic Oscillation (NAO) index (Jaagus and Kull, 2011). The strong reversal event and deep UML in winter 2011/12 were accompanied by an anomalously positive NAO index (Liblik et al., 2013). The mean December to February NAO index (Jones et al., 1997) in 2011/12 was 2.18. Likewise, the other three winters (2013/14, 2014/15, 2015/2016), when the mean UML depth in the gulf reached 60 m or deeper (Fig.

11a) had mean NAO index > 2 in December–February. Winters 2010/11 and 2012/13, which stand out in the time-series with re-stratification onset already in early January (Fig. 11a) had the lowest December–February averaged NAO indexes during the period 2010–2019: –1.06 and 0.47, respectively. Thus, large scale atmospheric forcing alters the re-stratification phenomena. Low NAO index conditions and easterly winds support re-stratification while high NAO index and westerly winds have the opposite effect.

It has to be noted that besides the frequency of easterly winds, also ice coverage tends to be larger in the case of low NAO (Jaagus, 2006). The landfast ice zone is expected to prevent vertical mixing and therefore supports lateral advection of riverine fresher water (Granskog et al., 2005). During the onset of re-stratification in late January 2012, the gulf was virtually ice-free. In both winters at the end of January, only the eastern part of the gulf and the adjacent part of the northern shore of the gulf were covered with ice. Thus, winter stratification

phenomena occurred even if most of the gulf was not covered by ice.

## 4. Discussion

  Positive net buoyancy flux is required for the onset of stratification in the upper layer. The processes
accounting for the negative buoyancy fluxes is vertical mixing caused by wind stirring, current shear, convection and vertical movements of the pycnocline. Positive buoyancy fluxes are resulting from the advection (arrival) of the lighter water in the sea surface or denser water to the subsurface. Likewise, warming of the surface layer at temperatures above $T_{md}$ or cooling below $T_{md}$ strengthens stratification. The positive buoyancy coming from the


cooling of the water below $T_{md}$ is rather small. If we consider salinity of 6 g kg$^{-1}$, the density difference between

waters at $T_{md}$ (2.8 °C) and freezing temperature (–0.3 °C) is 0.07 kg m$^{-3}$. This is the maximum density change if the water temperature is below $T_{md}$ and salinity is 6 g kg$^{-1}$. We get the same density difference if we keep temperature constant (1 °C) and vary salinity 0.09 g kg$^{-1}$. The changes in the sea surface salinity were approximately 1–2 g kg$^{-1}$ during winters (Fig. 8). Therefore, the effect of salinity change to the density and buoyancy flux was about 10–20-fold higher compared to temperature change in the gulf. We can conclude that

fresher water advection from the east was the primary source of buoyancy for the development of the stratification. This transport is controlled by wind forcing. Easterly winds support the fresher water advection to the west while westerly winds impede it (Liblik and Lips, 2012; Pavelson et al., 1997). In one hand, westerly winds generally deepen the mixed layer depth due to transport of the surface layer water from the northern Baltic Proper to the gulf (Liblik and Lips, 2017). On the other hand, if the fresher water is already present along the north coast, it spreads

to the south by westerly winds and creates stratification there (Figs. 4 and 8) and as noted in summer by Pavelson et al. (1997).

Wintertime stratification phenomena in nearshore regions, extending 10–20 km from the coast have been reported in several locations in the Baltic Sea (Granskog et al., 2005; Kari et al., 2018; Merkouriadi and Leppäranta, 2015). All these studies dealt with the phenomena under the ice. In the current study, on the example

of the Gulf of Finland, we showed that wintertime stratification could also occur in the basin-scale (along-gulf extent 400 km) and without considerable ice coverage. The western border of the phenomenon is around 23° E, i.e. at the entrance area to the gulf between Hiiumaa Island and the Finnish coast. Vertical mixing dominates over lateral buoyancy fluxes, and shallow stratification is not a common feature in the Baltic Proper. The occurrence of the shallow (<20 m) halocline reached over 50% in the northern part of the Gulf of Finland in February and the

southern part of the gulf in March. The high synoptic-scale and interannual variability of the upper mixed layer depth can be related to the wind regime and NAO index (Janssen et al., 2004), respectively. High wind stress, low ice cover, strong upwelling/downwelling (Janssen et al., 2004) and extreme estuarine circulation reversal events occur in the case of high NAO index (Liblik et al., 2013; Lilover et al., 2017; Lips et al., 2017; Suhhova et al., 2018). Enhanced vertical transport by upwelling/downwelling, wind stirring, and reversal events cause vertical

mixing, deepening of the UML, and upward transport of nutrients from the deeper layers (Janssen et al., 2004; Lilover and Stips, 2008; Lips et al., 2017). Low NAO index instead supports the consumption of riverine nutrients in the Gulf of Finland while the vertical mixing of nutrients from the deeper layer is modest.

We observed the Chl *a* concentration in the range 1.5–4.5 mg m$^{-3}$ in the first half of March in 2012 and 2014, i.e. comparable with the mean values in summer in the Gulf of Finland (Suikkanen et al., 2007). The higher

Chl *a* coincided with the cold and fresher upper layer and stronger stratification. Phytoplankton biomass (Fig. 12) generally follows Chl *a* structure (Fig. 6) in winter 2014. Observed biomass was in a similar order to typical summer values in the Baltic Sea (Bunse et al., 2016; Kudryavtseva et al., 2019).

Spring bloom evokes when the growth of the plankton exceeds losses in the upper layer due to grazing or vertical mixing downwards. Necessary conditions for the spring bloom are stabilized upper layer thinner than

euphotic zone, available nutrients and strong enough solar radiation. Upper mixed layer depth was 10–20 m in most of the gulf in early March 2012 and 2014. Euphotic layer depth estimated according to Luhtala and Tolvanen (2013) from our Secchi depth measurements was 15–19 m in both winters, i.e. comparable with the UML depth.





Also, there were enough nutrients available in the upper layer in 2014 February and March (Lips et al., 2017). We
do not have a reference for the nutrients data in 2012. Thus, the limiting factor for phytoplankton growth is likely

insufficient solar radiation. The mean downward shortwave radiation doubles in the area from February (40–50
$Wm^{-2}$) to March (90–100 $Wm^{-2}$) and quadruples in April (160–200 W $m^{-2}$) (Rozwadowska and Isemer, 1998;
Zapadka et al., 2020). The onset of spring bloom typically occurs in April in the gulf (Groetsch et al., 2016; Lips
et al., 2014; Lips and Lips, 2017).


## 4. Conclusions

We have demonstrated by in-situ measurements and numerical modelling that haline stratification at the
depth comparable to the euphotic zone depth occurs in the Gulf of Finland during wintertime, well before the onset

of thermal stratification in spring. Stratification forms in late January–early February as a result of the advection
of riverine water to west along the northern coast of the gulf. Stratification is maintained by the positive buoyancy
flux created by the advection, which is stronger than the negative flux resulting from vertical mixing. Fresher water
and haline stratification appeared approximately one month later along the southern coast of the gulf. Earlier
observations of a local stratification phenomenon in the Baltic Sea in winter have been registered in ice-coverage

conditions. Our observations showed that haline stratification could occur in the whole gulf and without ice cover.
Therefore we can assume that wintertime stratification is a common phenomenon in the Gulf of Finland. Its
western boundary is at the entrance area to the gulf, between Hiiumaa Island and the Finnish coast.

Elevated Chl *a* and phytoplankton biomass were registered in the mixed layer above the halocline. The
limiting factor for phytoplankton growth in winter was likely insufficient light radiation. The exact role of

wintertime stratification in the nutrient cycle and plankton dynamics in the Gulf of Finland needs further
investigations.

*Code availability.* Scripts to analyze the results are available upon request. Please contact TL.

*Author contributions.* TL led the analyses of the data and writing of the manuscript with contributions of GV, JL,
M-JL and IL. TL was responsible for the measurements and GV for the modelling activities. VK was responsible
for gathering and processing of the Ferrybox data. IL arranged fytoplankton biomass measurements.

*Competing interests.* We declare that no competing interests are present.


*Acknowledgments.* This work was financially supported by the Estonian Research Council grant (PRG602) and
Institutional Research Funding IUT (IUT19-6) and Estonian Science Foundation grant 9382. We thank our
colleagues and the crew of R/V Salme in the field works. Likewise, we are thankful to Tallink (Estonia) for the
possibility to acquire measurements on ferries. We thank U. Lips for the arrangements of RV Salme cruises and

Ferrybox measurements, T. Kõuts for providing Tallinnamadal wind data.



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



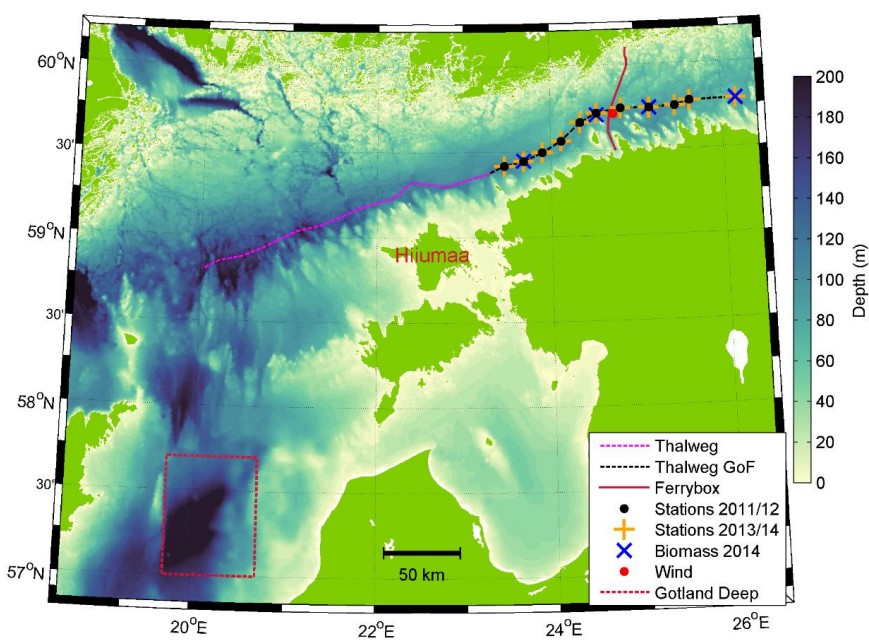


**Fig. 1. Bathymetric map of the Baltic Proper and the Gulf of Finland. Thalweg from the Central Gulf of Finland, CTD stations visited in 2011/12 and 2013/14, phytoplankton biomass sampling stations, Tallinn–Helsinki Ferrybox line, Tallinnamadal wind measurements location and Gotland Deep area are shown.**

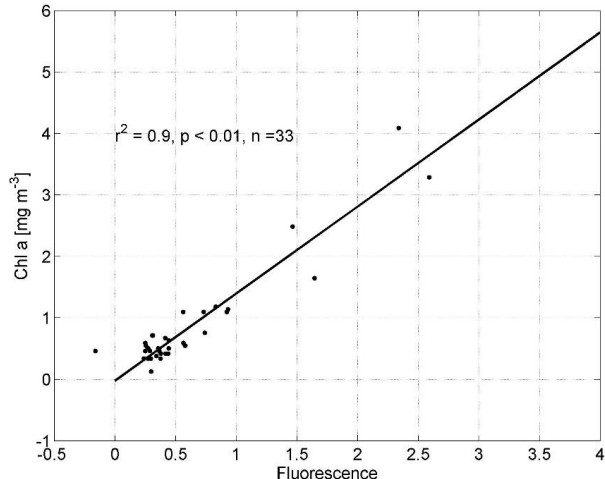

**Fig. 2. Chl *a* fluorescence vs Chl *a* (mg m$^{-3}$) measured from water samples.**





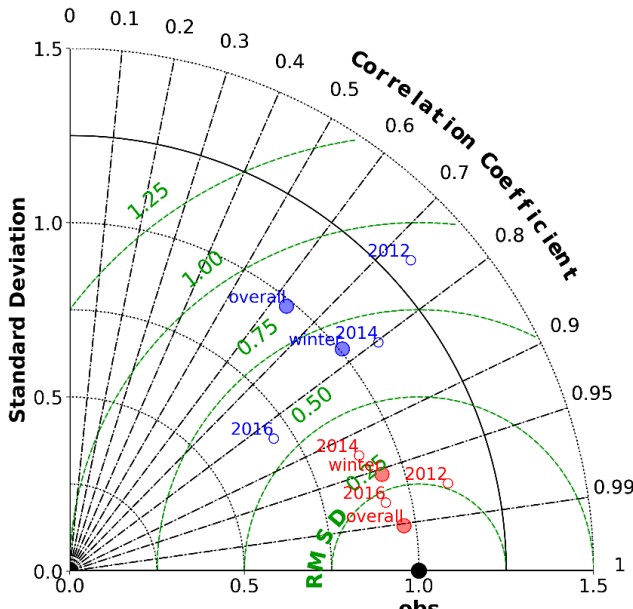

**Fig. 3. Taylor diagram of simulated and measured temperature (red) and salinity (blue) along the Ferrybox transect from Tallinn to Helsinki. Overall – all the available observations from 1.11. 2011 to 1.06. 2016 (filled circles), winter – all the available observations from December to the end of March in 2011–2016 (filled circles), and winter observations from January–March in 2012, 2014 and 2016 (circles).**



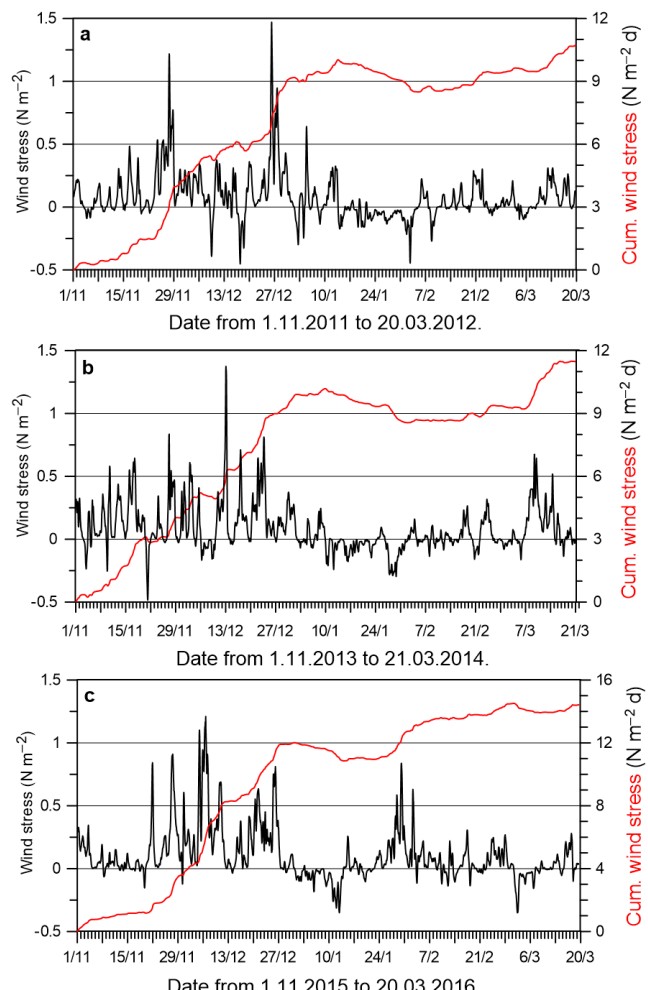

**Fig. 4. Time-series of an along-gulf component of wind stress (black curve, positive eastward) and cumulative along-gulf wind stress (red curve) based on wind data from 1 November 2011 to 20 March 2012 (a), from 1 November 2013 to 21 March 2014 (b) and from 1 November 2011 to 20 March 2016 measured at**
**Tallinnamadal Lighthouse in the Gulf of Finland.**

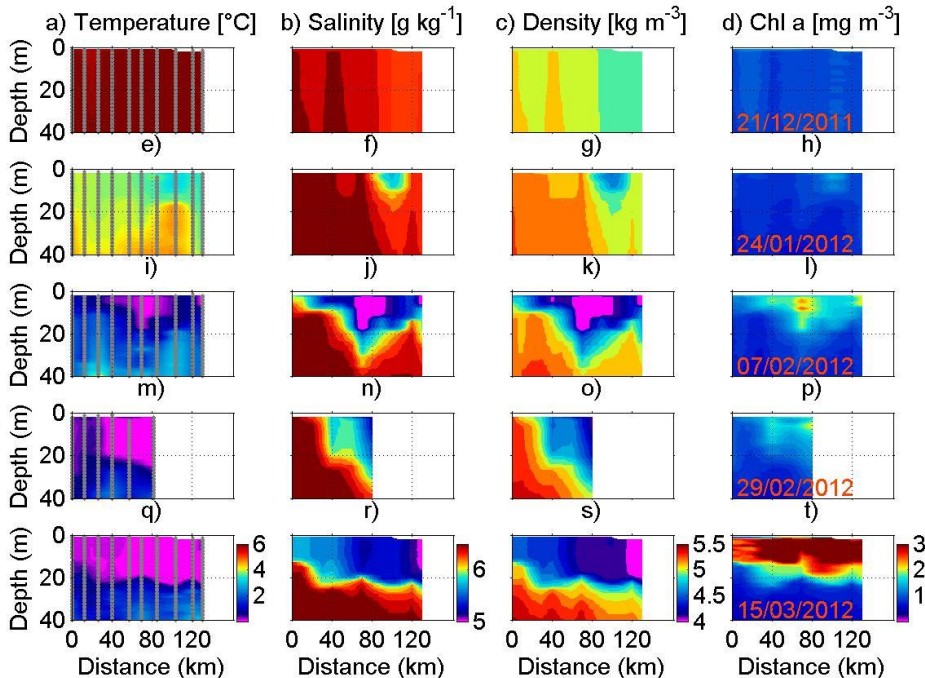

**Fig. 5. Vertical sections of temperature, salinity, density anomaly, and Chl *a* on 21 December 2011, 24–25 January 2012, 7–8 February, 29 February and 15–16 March along the Gulf of Finland (black dots in Fig1) from west to east. Grey lines show the location of CTD-casts.**


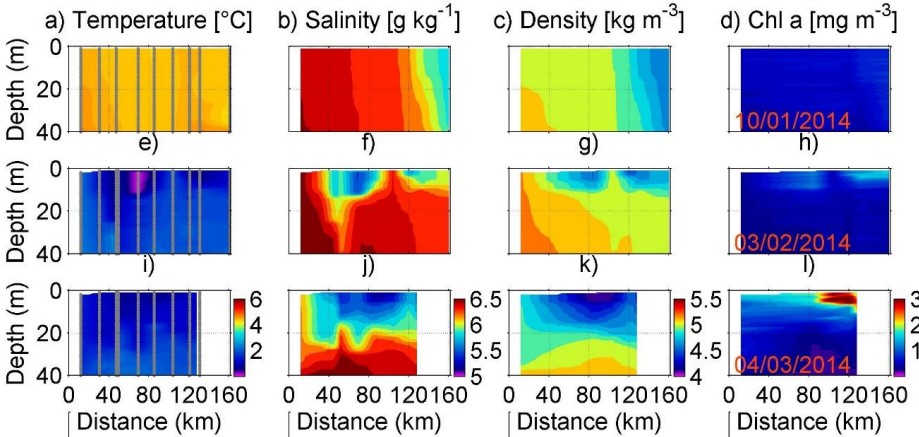

**Fig. 6. Vertical sections of temperature, salinity, density anomaly and Chl *a* on 9–10 January, 3–4 February, and 4–5 March 2014 along the Gulf of Finland (orange crosses in Fig1) from west to east. Grey lines show the location of CTD-casts.**




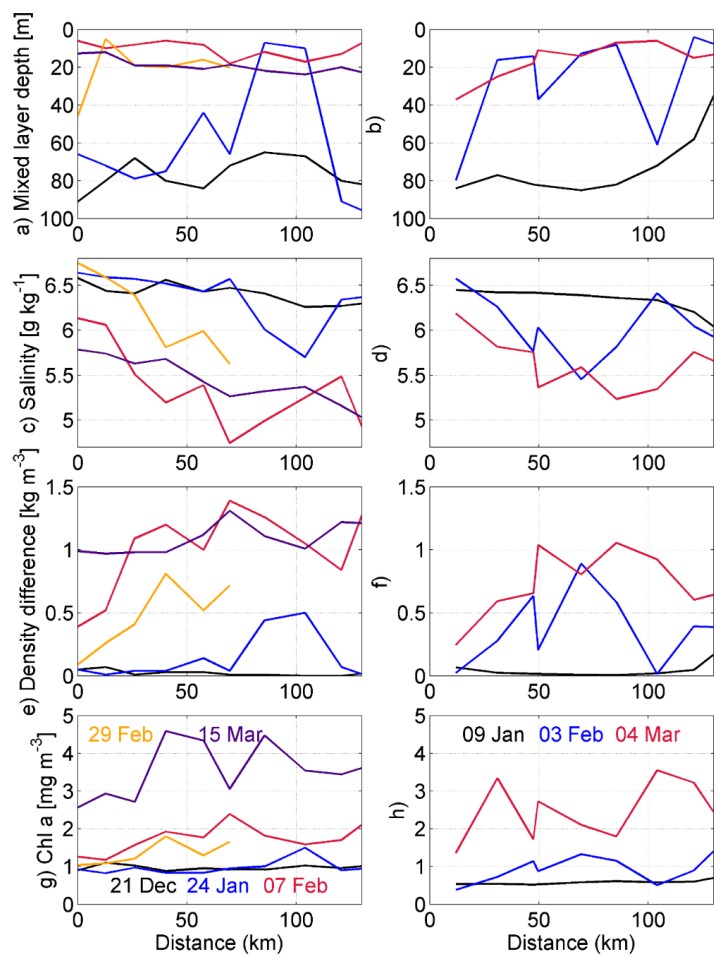

**Fig. 7. (a) Mixed layer depth, (b) salinity in the upper 5 m, (c) density difference between the surface layer and 40 m depth and (d) Chl *a* concentration in the upper 5 m along the Gulf of Finland from west to east in winters 2011/12 (left panels) and 2013/14 (right panels). Dates of measurements are shown on the lower panel, and colours indicate the corresponding lines.**






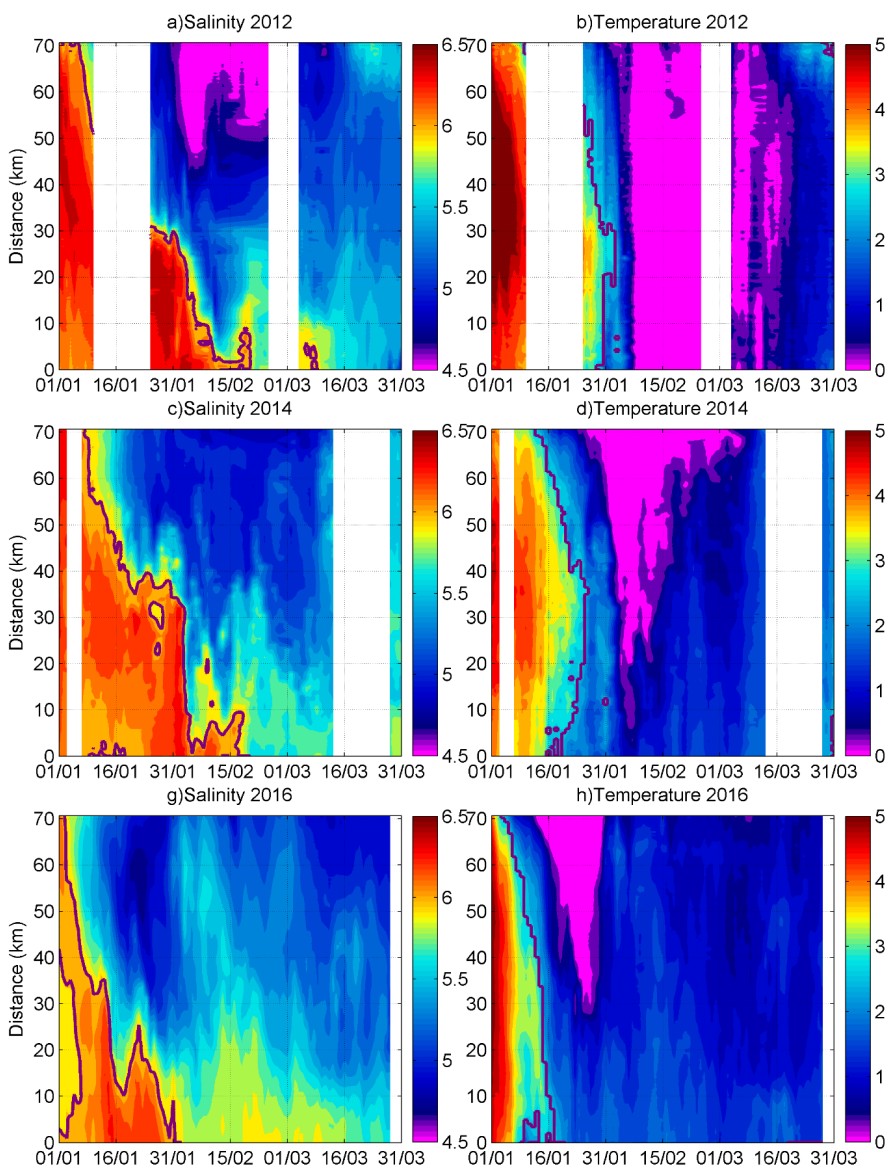

Fig. 8. Salinity (g kg⁻¹) and temperature (°C) in the upper layer along the transect from Tallinn to Helsinki (red line in Fig. 1) January–March 2012, 2014 and 2016. Isoline 6 g kg⁻¹ is shown on salinity plot and maximum density temperature $T_{md}$ on temperature plot.




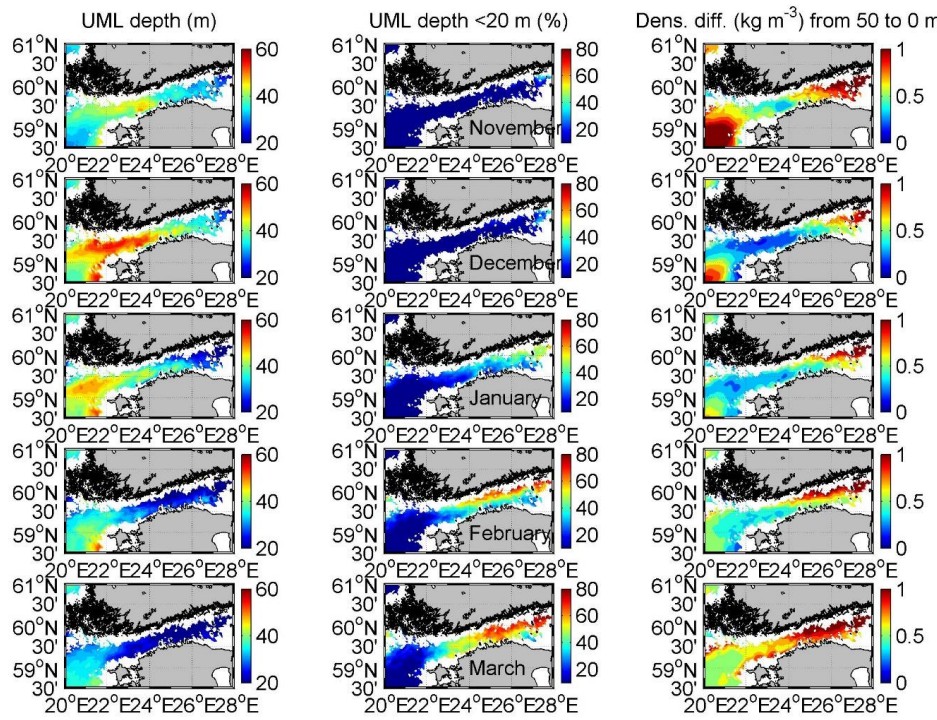


**Fig. 9. Mean simulated upper mixed layer (UML) depth and occurrence of the UML depth <20 m from November (uppermost panel) to March (lowermost panel) from 2010 to 2019.**

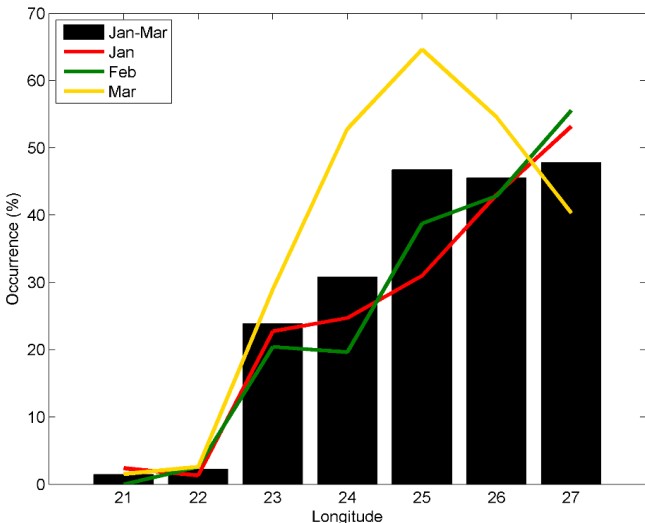


**Fig. 10. The occurrence of the density difference >0.5 kg m⁻³ between 40 m depth and sea surface in the Gulf of Finland from January to March in 1904–2020. In total 2560 temperature-salinity data pairs in the surface layer and at 40 m depth were included.**




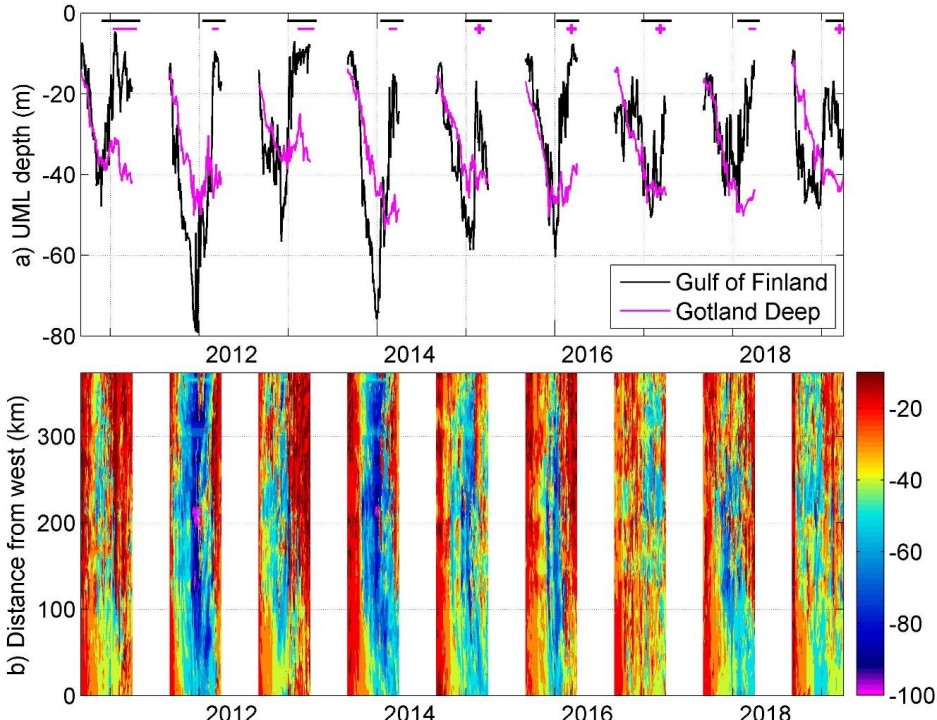

**Fig. 11. (a) the mean upper mixed layer depth time-series in the Gulf of Finland and Gotland Deep based on model simulation data from autumn 2010 to spring 2019 and (b) upper mixed layer depth along the transect from the northern Baltic Proper to the Central Gulf of Finland (see Thalweg in Fig. 1). Only periods from October to March are shown. Horizontal lines in the upper panel show the periods when remotely sensed SST was below $T_{md}$, and crosses indicate the day of minimum temperature in winters when the temperature below $T_{md}$ was not observed in the Gotland Deep. The areal mean in plot a is calculated for the transect in the Gulf of Finland (Thalweg GoF, Fig. 1) and the box in the Gotland Deep (Fig. 1).**





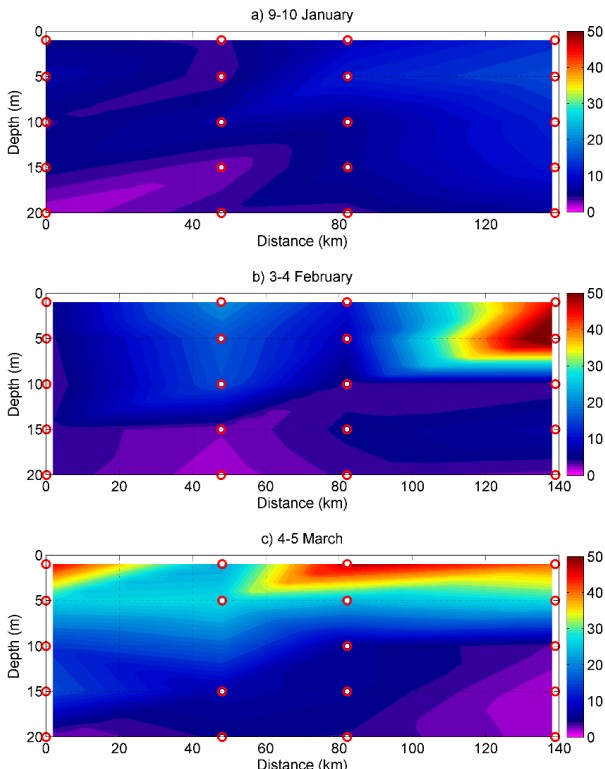


**Fig. 12. Vertical distribution of phytoplankton carbon content (mgC m⁻³) along the Gulf of Finland in winter 2014. Red circles show the locations of sampling (blue crosses in Fig. 1).**