# Peer review of "Winter stratification phenomenon and its consequences in the Gulf of Finland, Baltic Sea"

_Ocean Science, 2020_

## Referee Comment (RC1) · Anonymous Referee #1 · 5 Jul 2020

**General comments**

This manuscript documents the formation of wintertime haline stratification in the Gulf of Finland due to freshwater transport and discusses its implications for early plankton bloom dynamics. The authors combined water column temperature, salinity and fluorescence data from two along-Gulf transects in winters 2011–2012 and 2014, cross-Gulf measurements of surface T-S collected with a Ferrybox system and 10 years of GETM-modelled mixed-layer depths. Altogether this is a powerful dataset that allowed to a thorough documentation and description of an interesting phenomenon, which has implications for (usually disregarded) winter primary production in the area. Overall, my view on the manuscript is quite positive and I would be happy to see it published. Still,

there are several formal issues that need to be addressed before publication. I also feel that the description of some aspects of the dynamics of the system could be described in more depth. I develop this further below but, for example, the general seasonal wind patterns in the area, and how they relate to expected advection patterns are poorly discussed in my view. The authors have nice model simulations and a set of references to better describe the advection dynamics of the system in response to changing winds. I would suggest to develop this aspect a bit more. I feel that if the authors could condensate this information together with their own conclusions in a schematic figure that would help a lot and make the manuscript more shinny and visual.

**Specific comments**

**Line 179.** Be careful with the positioning of parenthesis for references.

**Wind pattern.** The wind pattern (Fig. 6) is strikingly similar for the three years shown here, with strong westerly winds until January and weaker more variable winds after that. Is this the typical seasonal pattern in the region? I think this is a very important point for your message that is not very well developed in the manuscript. You focus more on interannual variation and the links to NAO, but what are the expected seasonal variations of wind forcing during the studied period. Is this transition from strong westerlies to weak variable winds over winter a persistent pattern? Then this is very important for the onset of wintertime stratification. Could you develop this a bit more please?

**Figure 7 and lines 212-227** I like Figure 7, it is quite illustrative, but the only really new information displayed in this figure is the mixed-layer depth. Consequently, some of the information provided in lines 212-227 becomes somewhat repetitive. As the paper has a long number of display items I would suggest to show the MLD already in Figure 6.

**Figure 8.** Could you highlight in the caption the location of the starting point of the transects (x = 0 km)?

**Line 237** "Spreading from the east to west". This information is not really contained in the Figure. In my view it is a bit confusing to include it in the middle of this sentence which is, otherwise, a pure description of the information that is being displayed.

**Line 245** Which year are you talking about? Also I am curious about the fact that the onset of haline and, more importantly, thermal stratification seems to have taken place early than in the previous years. Is this related to variability in wind forcing?

**Figure 11** The x-labels are placed in a strange way in this figure. Do the ticks correspond to the 1st of January of each year? Why is the label to the right of the tick? The color scale for MLD in panel b) is reversed with respect to Figure 10. This confused me.

**Lines 280-294.** I think this part is very interesting but needs to be improved. From Figure 11 it is a bit hard to compare the timing of stratification on-set in the different years. I would try to rethink this figure a bit and find a better way to make your point. Also the winter NAO index is an important element here. I would add this information to the figure somehow.

**Line 305.** In my view "vertical movements of the pycnocline" due to upwelling, internal waves, etc, are transient and have a mostly reversible effect on buoyancy fluxes unless part of their energy is irreversibly lost to turbulent mixing. I would avoid mentioning them or explain better what you mean.

**Lines 326-328.** "The western border of the phenomenon is around 23°E, i.e. at the entrance area to the gulf between Hiiumaa Island and the Finnish coast. Vertical mixing dominates over lateral buoyancy fluxes, and shallow stratification is not a

common feature in the Baltic Proper." I find this quite sharp boundary intriguing. Could you add some reference for this or develop a bit more this subject? Why is this change in regime, is the Baltic proper much more wind exposed so that haline stratification is completely eroded? Or is it that some dynamical process precludes the advection of freshwater out of the Gulf?

**Lines 343–345.** This sentence needs a reference.

**Figure 12.** I don't like this figure very much. There is very few data available for interpolation. Why don't you use a scatter plot of biomass (with a color/size code) superimposed to a salinity contour plot? This would maybe make your point stronger.

**Color scale.** In the contour figures you use a highly non linear colormap which strengthens low values a lot. I feel that sometimes the use of such a colorscale can be misleading, as it attracts the attention of the reader to this very low values, and sometimes this is not the most relevant aspect. I would suggest that the authors re-think a bit this choice for certain figures.

---

## Referee Comment (RC2) · Daniel Carlson (Referee) · 5 Jul 2020

In fairness to the authors, the editorial staff should not have requested reviews for a manuscript in such a rough state. I think it could warrant publication at some point, but as it is, the manuscript is not even ready for submission. Overall, the manuscript reports observational and model results but it fails to put these results into context and it fails to provide any motivation for the study. The science appears to be sound, but it's not clear why it was done. The manuscript requires extensive editing for grammar and style. There are too many minor grammatical errors for me to keep track of. Tell the reader early on why your work matters and how it fits into a larger context. I am sure it is important, but as it is written now, the manuscript fails to convey that importance. I recommend reading Mensh and Kording (2017) Ten simple rules for structuring papers.

[Figure]

PLOS Computational Biology 10.1371/journal.pcbi.1005619

Given that the manuscript lacks a clear motivation, it is difficult to evaluate it using the journals review criteria, which are listed here: https://www.ocean-science.net/peer_review/review_criteria.html

The authors may also wish to use these criteria when revising their manuscript.

+ 'Phenomena' is typically the plural form of phenomenon (https://www.merriam-webster.com/dictionary/phenomenon) So the title should read '... phenomena and their consequences . . .' or 'Winter stratification and its consequences...'

+ Introduction- the first paragraph of the introduction should state the main goal or problem that the manuscript aims to address. As it is written now, the first paragraph is full of many details about the seasonal cycle of stratification in the Baltic Sea but we are left guessing as to the importance of these details. Please tell the reader the main point or what particular issue your manuscript addresses in the first few sentences and then move on to specific information that the reader needs to understand what has been done, and what is new.

+ The motivation for this manuscript is not stated until line 75 - "Details about the formation of the haline stratification in the larger areas of the Baltic Sea during wintertime is mainly unknown." This is the new topic that you address. Please make that clear in the first paragraph and then tell us about what is known. When it's the other way around, we're left wondering why you are telling us all this information and where it is going.

+ The OSTIA product is not technically remote sensing data. It is gap-filled remote sensing data that also uses in situ observations. The most recent citation for OSTIA is Good et al. (2020) The Current Configuration of the OSTIA System for Operational Production of Foundation Sea Surface Temperature and Ice Concentration Analyses. Remote Sensing. 12:720 doi:10.3390/rs12040720

[Figure]

+ Regarding the OSTIA data- from the text it's not clear if you used daily OSTIA fields or mean SST for the entire period from 2010-2019. Can you please clarify?

+ Section 1.2- why use nautical miles? The journal requires the use of metric units. https://www.ocean-science.net/for_authors/manuscript_preparation.html

+ What are your open boundary conditions? Relaxation? How is riverine input treated in the model? How did you spin up the model?

+ Results- since the response of the water column was very similar in each wind-driven event, perhaps describe the general behavior first - strong winds, well-mixed water column, low chl-a.

+ Line 219 - "Since freshwater originates from the east..." Is this statement supported by data or is it speculation? If it's speculation, please move speculative arguments to the Discussion.

+ Instead of describing what was observed in each dataset, use all the data to describe the stratification phenomena of interest. You are using widely accepted methods so there is no need to justify their use. Communicate your point clearly and succinctly

+ Line 295 - this also appears to be speculative and should be moved to the discussion

+ Figure 2 is not necessary. Simply state the r^2, p value, and n in the manuscript

+ Figure 5 contains too many subplots. It's cluttered and difficult to take in

+ Figure 9 also contains too many subplots. Perhaps create an animation?

---

## Author Response (AR1)

RESPONSES TO REVIEWERS

**Reviewer 1**

**Comment:** This manuscript documents the formation of wintertime haline stratification in the Gulf of Finland due to freshwater transport and discusses its implications for early plankton bloom dynamics. The authors combined water column temperature, salinity and fluorescence data from two along-Gulf transects in winters 2011–2012 and 2014, cross Gulf measurements of surface T-S collected with a Ferrybox system and 10 years of GETM-modelled mixed-layer depths. Altogether this is a powerful dataset that allowed to a thorough documentation and description of an interesting phenomenon, which has implications for (usually disregarded) winter primary production in the area. Overall, my view on the manuscript is quite positive and I would be happy to see it published. Still there are several formal issues that need to be addressed before publication. I also feel that the description of some aspects of the dynamics of the system could be described in more depth. I develop this further below but, for example, the general seasonal wind patterns in the area, and how they relate to expected advection patterns are poorly discussed in my view. The authors have nice model simulations and a set of references to better describe the advection dynamics of the system in response to changing winds. I would suggest to develop this aspect a bit more. I feel that if the authors could condensate this information together with their own conclusions in a schematic figure that would help a lot and make the manuscript more shinny and visual.
**Reply:** Thank you for your review and helpful comments! We have addressed all of the points you have highlighted. We have added new schematic figure (12).
**Action:** As we explain below, the seasonal wind pattern and its role is now dealt in the paper. We added thematic figure (last one), which explains the UML distributions in the case of westerly wind and easterly wind dominance. We have addressed all the detailed comments below.

**Comment:** Line 179. Be careful with the positioning of parenthesis for references.
**Reply:** We fixed it.
**Action:** Done

**Comment:** Wind pattern. The wind pattern (Fig. 6) is strikingly similar for the three years shown here, with strong westerly winds until January and weaker more variable winds after that. Is this the typical seasonal pattern in the region? I think this is a very important point for your message that is not very well developed in the manuscript. You focus more on interannual variation and the links to NAO, but what are the expected seasonal variations of wind forcing during the studied period. Is this transition from strong westerlies to weak variable winds over winter a persistent pattern? Then this is very important for the onset of wintertime stratification. Could you develop this a bit more please?
**Reply:** Yes, we agree. This is a very good point you made. Thank you! Yes, it is part of the annual cycle of wind. The cycle is not as persistent as it looks from the three selected years, but it exists. There is a period from October/November to January when there are more westerly storms and after that, when atmospheric high pressure systems sustain, it calms down. The timing and magnitude vary from year to year though.

**Action:** We have added a new figure showing the annual cycle of along gulf wind stress, we have added sentences about it to the 3.2 (results), discussion, conclusions, and abstract.

**Comment:** Figure 7 and lines 212-227 I like Figure 7, it is quite illustrative, but the only really new information displayed in this figure is the mixed-layer depth. Consequently, some of the information provided in lines 212-227 becomes somewhat repetitive. As the paper has a long number of display items I would suggest to show the MLD already in Figure 6.
**Reply:** Yes, we agree with the comment.
**Action:** We removed the text in 212-227 and added MLD to the figures 5 and 6. We made three minor changes in the previous two sections, just to mention the upper mixed layer depth there. Otherwise, we think the section was repetitive, as you noted.

**Comment:** Figure 8. Could you highlight in the caption the location of the starting point of the transects (x = 0 km)?
**Reply:** Yes, that is a good idea.
**Action:** We added, "the starting point of the transect (x = 0 km) is in the Bay of Tallinn at 59.500° N and 24.752° E."

**Comment:** Line 237 "Spreading from the east to west". This information is not really contained in the Figure. In my view it is a bit confusing to include it in the middle of this sentence which is, otherwise, a pure description of the information that is being displayed.
**Reply:** Yes, we agree.
**Action:** We removed "Spreading from the east to west", but added next sentence to explain the freshwater origin: "Since the main sources of freshwater are in the east, water must have flown westward along the northern coast."

**Comment:** Line 245 Which year are you talking about? Also I am curious about the fact that the onset of haline and, more importantly, thermal stratification seems to have taken place early than in the previous years. Is this related to variability in wind forcing?
**Reply:** We talked about 2014 and 2016. We added an explaining sentence. The earlier onset in 2016 is related to the wind forcing. One can see it in figure 4c. The westerlies eased off earlier in 2016 than two other years. We added a sentence about it to the manuscript.
**Action:** we changed "The onset of haline stratification occurred slightly earlier in 2016 due to wind forcing – the westerlies had eased off by the end of December 2015 (Fig. 3c)."

**Comment:** Figure 11 The x-labels are placed in a strange way in this figure. Do the ticks correspond to the 1st of January of each year? Why is the label to the right of the tick? The color scale for MLD in panel b) is reversed with respect to Figure 10. This confused me.
**Reply:** We agree and fixed the issues.
**Action:** We solved the problem with ticks and we put the color scale the same way as in the previous figure.

**Comment:** Lines 280-294. I think this part is very interesting but needs to be improved. From Figure 11 it is a bit hard to compare the timing of stratification on-set in the different years. I would try to rethink this figure a bit and find a better way to make your point. Also the winter NAO index is an important element here. I would add this information to the figure somehow.

**Reply:** Thank you for this recommendation.
**Action:** We added another subplot, where one can see detailed time-series of UML in different years. We also added the Dec-Feb mean NAO index to the second subplot.

**Comment:** Line 305. In my view "vertical movements of the pycnocline" due to upwelling, internal waves, etc, are transient and have a mostly reversible effect on buoyancy fluxes unless part of their energy is irreversibly lost to turbulent mixing. I would avoid mentioning them or explain better what you mean.
**Reply:** We agree.
**Action:** We removed "vertical movements of the pycnocline".

**Comment:** Lines 326-328. "The western border of the phenomenon is around 23∘E, i.e. at the entrance area to the gulf between Hiiumaa Island and the Finnish coast. Vertical mixing dominates over lateral buoyancy fluxes, and shallow stratification is not a common feature in the Baltic Proper." I find this quite sharp boundary intriguing. Could you add some reference for this or develop a bit more this subject? Why is this change in regime, is the Baltic proper much more wind exposed so that haline stratification is completely eroded? Or is it that some dynamical process precludes the advection of freshwater out of the Gulf?
**Reply:** Yes, this can be mentioned here. We think the feature can occur in the Gulf of Finland because of the two factors: the high riverine input and elongated shape. The phenomenon vanishes in the area, where the extension of the gulf (at the entrance of the gulf) gets wider. Likewise, it is simply far from the main freshwater sources.
**Action:** We added the following text to the manuscript: "The absence of the phenomenon in the Baltic Proper can be explained by the long distance from rivers, due to its larger size and topography. Riverine input per unit area in the Gulf of Finland is 7–8 times larger than in the Baltic Proper (Leppäranta and Myrberg, 2009). As the wintertime stratification phenomenon vanishes at the wider entrance area to the Gulf of Finland, it is likely that the elongated, narrow shape of the gulf accounts contributes to the formation of stratification as well as high freshwater input."

**Comment:** Lines 343–345. This sentence needs a reference.
**Reply:** We agree.
**Action:** We added Smetacek and Passow (1990) and Fennel (1990).

**Comment:** Figure 12. I don't like this figure very much. There is very few data available for interpolation. Why don't you use a scatter plot of biomass (with a color/size code) superimposed to a salinity contour plot? This would maybe make your point stronger.
**Reply:** We agree, this figure can be better designed.
**Action:** We have changed the figure according to your recommendation.

**Comment:** Color scale. In the contour figures you use a highly non linear colormap which strengthens low values a lot. I feel that sometimes the use of such a colorscale can be misleading, as it attracts the attention of the reader to this very low values, and sometimes this is not the most relevant aspect. I would suggest that the authors re-think a bit this choice for certain figures.
**Reply:** We agree.
**Action:** We have changed the color scales of figs. 5, 6, 11, 12 (according to the first submission numbering).

**Reviewer 2.**

**Comment:** In fairness to the authors, the editorial staff should not have requested reviews for a manuscript in such a rough state. I think it could warrant publication at some point, but as it is, the manuscript is not even ready for submission. Overall, the manuscript reports observational and model results but it fails to put these results into context and it fails to provide any motivation for the study. The science appears to be sound, but it's not clear why it was done. The manuscript requires extensive editing for grammar and style. There are too many minor grammatical errors for me to keep track of. Tell the reader early on why your work matters and how it fits into a larger context. I am sure it is important, but as it is written now, the manuscript fails to convey that importance. I recommend reading Mensh and Kording (2017) Ten simple rules for structuring papers. C1 -PLOS Computational Biology 10.1371/journal.pcbi.1005619
**Reply:** Thank you for your comments and recommendation.
**Action:** We wrote about the motivation at lines 28-36 and 75-83. To provide for a reader the importance of the paper in a larger context, we added a section to the beginning of the manuscript. The results chapter was shortened to keep it more condensed, introductive sentences were added to each section, to make it easier to read. We arranged extensive editing for grammar and style.

**Comment:** Given that the manuscript lacks a clear motivation, it is difficult to evaluate it using the journals review criteria, which are listed here:
https://www.oceanscience.net/peer_review/review_criteria.html. The authors may also wish to use these criteria when revising their manuscript.
**Reply:** We have added a section to the very beginning of the introduction to indicate what is the overall motivation of the study. Details about motivation (why stratification is important for the physics, biogeochemistry and biology) were given at lines 28-36 (previous submission).
**Action:** We added a section to the beginning of the section. We checked the criteria before revising the manuscript.

**Comment:** + 'Phenomena' is typically the plural form of phenomenon
(https://www.merriamwebster. com/dictionary/phenomenon) So the title should read ': : : phenomena and their consequences : : :' or 'Winter stratification and its consequences'
**Reply:** Thank you for the note.
**Action:** We changed it to "phenomenon".

**Comment:** + Introduction- the first paragraph of the introduction should state the main goal or problem that the manuscript aims to address. As it is written now, the first paragraph is full of many details about the seasonal cycle of stratification in the Baltic Sea but we are left guessing as to the importance of these details. Please tell the reader the main point or what particular issue your manuscript addresses in the first few sentences and then move on to specific information that the reader needs to understand what has
been done, and what is new.
**Reply:** Yes, this paragraph can be added in the beginning.
**Action:** We added: "Upper layer stratification is an important characteristic in the dynamics of the pelagic ecosystem. However, to our knowledge, the formation of wintertime haline stratification in the upper layer of the whole Gulf of Finland has not been investigated; the present study focuses

on the formation of wintertime haline stratification caused by freshwater inflow and wind forced circulation, and the observed haline stratification explains early phytoplankton dynamics."

**Comment:** + The motivation for this manuscript is not stated until line 75 - "Details about the formation of the haline stratification in the larger areas of the Baltic Sea during wintertime is mainly unknown." This is the new topic that you address. Please make that clear in the first paragraph and then tell us about what is known. When it's the other way around, we're left wondering why you are telling us all this information and where it is going.
**Reply:** Yes, it can be mentioned earlier.
**Action:** Please see our previous response.

**Comment:** + The OSTIA product is not technically remote sensing data. It is gap-filled remote sensing data that also uses in situ observations. The most recent citation for OSTIA is Good et al. (2020) The Current Configuration of the OSTIA System for Operational Production of Foundation Sea Surface Temperature and Ice Concentration Analyses. Remote Sensing. 12:720 doi:10.3390/rs12040720
**Reply:** Thank you for the information.
**Action:** We removed "remote sensing" from the text in several places and added the reference.

**Comment:** + Regarding the OSTIA data- from the text it's not clear if you used daily OSTIA fields or mean SST for the entire period from 2010-2019. Can you please clarify?
**Reply:** We use daily OSTIA fields. We clarified this in the data and methods part.
**Action:** It reads now "OSTIA (Donlon et al., 2012; Good et al., 2020) daily mean sea surface temperature (SST) data for the period 2010–2019 were obtained from the Copernicus Marine Environment Monitoring Service."

**Comment:** + Section 1.2- why use nautical miles? The journal requires the use of metric units. https://www.ocean-science.net/for_authors/manuscript_preparation.html
**Reply:** Metric units for the GETM run are and were also written in the text. We used nautical miles as those are also commonly used in the modelling community – the original grid is planned in nautical miles (either 1 n.m. or 2 n.m., in our case 0.5 n.m.) and it makes sense to indicate this. See e.g. https://os.copernicus.org/articles/15/1691/2019/ or https://os.copernicus.org/articles/15/1399/2019/
**Action:** We added metric units also for the COPERNICUS reanalysis product.

**Comment:** + What are your open boundary conditions? Relaxation? How is riverine input treated in the model? How did you spin up the model?
**Reply:** Thank you for the remark, we have made adjustments in the model description. The boundary conditions for different parameters differ in the model. Observed sea surface height is set to the boundary with Flather (1994) radiation scheme, while the temperature and salinity are relaxed to climatological profiles (Janssen et al. 1999 along the open boundary. There is also a sponge layer with 3-points for the latter.

Riverine water has a constant salinity 0.5 g/kg due to numerical reasons. We are pretty sure that the model can handle also lower values, but at least in the used experiment, it was 0.5. As the

temperatures for different rivers are not known/hard to prescribe, the model uses target cell value. Riverine water enters as a change in the sea surface height – the volume of entered water within one model iteration divided by target cell area will give the additional change in the SSH.

We use re-analysis product from Copernicus Marine Service as the initial temperature and salinity. By definition, it is supposed to be the best available possibility to get 3D field of T/S as it "interpolates" observations using the state-of-the-art method. In reality, one can always argue whether it is the "best" way. Nevertheless, we assume that the re-analysis product has all correct salinities and temperatures for different basins and is already baroclinically balanced. Our simulations start from the motionless state but as the Baltic Sea is shallow and wind-driven circulation prevails, the model will quickly adjust to forcing. Lips et al (2016) showed that the volume-averaged kinetic energy reaches correct values within 5-days. In summary, we do not think further spin-up is necessary for the simulations.

**Action:** We have considerably modified the section 1.2.

**Comment:** + Results- since the response of the water column was very similar in each winddriven event, perhaps describe the general behavior first - strong winds, well-mixed water column, low chl-a.
**Reply:** We considered this but realized that it is better if this comes after we have described the observations. Strong westerlies- well mixed water column relation is given in the introduction and the possibility of the formation of the shallow halocline is also mentioned there.
**Action:** No particulate action here, but we think with other changes (complementing introduction, and introducing each section in the results chapter) help a reader to follow the results.

**Comment:** + Line 219 - "Since freshwater originates from the east: : :" Is this statement supported by data or is it speculation? If it's speculation, please move speculative arguments to the Discussion.
**Reply:** We expected here a reader noted from the introduction that the riverine water enters the gulf mainly from the eastern part of the gulf. Anyhow according to the other reviewer comment we realized this section is repeating the previous section, so we decide to remove it.
**Action:** We removed this section.

**Comment:** + Instead of describing what was observed in each dataset, use all the data to describe the stratification phenomena of interest. You are using widely accepted methods so there is no need to justify their use. Communicate your point clearly and succinctly
**Reply:** We describe the phenomenon in the order of subtopics. In the first two sections, we describe the stratification phenomenon along the gulf, its vertical structure and impact on Chl *a*. Next, we describe the surface characteristics (measured by the ferrybox two times a day), which give us a more detailed understanding of the temporal developments of haline stratification formation. In 3.2 we make statistics of the process based on the model and historical CTD data to put our results to a broader context. We added introducing sentences before sections and renamed chapter 3.1. to make it easier

**Action:** We renamed the 3.1 title and added introducing sentences to each section to make it easier to follow for a reader. Likewise, we have shortened 3.1 considerably. We also removed figure 7 after comment from another reviewer.

**Comment:** + Line 295 - this also appears to be speculative and should be moved to the discussion
**Reply:** We change the first sentence of this section to make it less speculative. There is a strong correlation between ice coverage and the NAO index in winter according to literature. But we agree, since the section includes references to previous studies, it rather belongs to the discussion.
**Action:** We moved the section to the discussion.

**Comment:** + Figure 2 is not necessary. Simply state the r^2, p value, and n in the manuscript
**Reply:** We agree.
**Action:** We removed the figure and added a sentence about it to the manuscript.

**Comment:** + Figure 5 contains too many subplots. It's cluttered and difficult to take in
**Reply:** We agree, this figure needed improvement.
**Action:** We remade figure 5. We hope it looks better now.

**Comment:** + Figure 9 also contains too many subplots. Perhaps create an animation?

**Reply:** We agree, this figure needed improvement. We believe it is better to keep it in figure format though.

**Action:** We modified the figure. The axis labels were removed (except in one plot) and the plots are now more zoomed in to the area of interest. We believe after these changes the figure is much easier to read.

Manuscript with track changes

[revised manuscript text omitted]
 in winter 2016 (January–March) and larger for in 2012, while for in 2014 it is close to the observedobservations. The simulated variability of the temperature is captured well —  the standard deviations from the simulations are at least 0.8 of the observed for all the time periods, although the . The model slightly overestimated the temperature variability for the winter of 2012.

For salinTity, the overall correlation coefficient for salinity is 0.62, while it is over 0.74 both for both the whole wintertime period and single years as well. There is a A higher correlation for temperature (as expected); o is for temperature. Overall correlation, which includes as the seasonal variability, is included, is 0.99, and while for the wintertime it is 0.95. Very high correlation (>0.94) for the temperature is also shown for single individual winters.

The Root mean root-squared differences between model and observed values are slightly larger for the salinity but do not exceed the observed variability. In general, the model captures the wintertime changes in the surface layers of the Gulf of Finland well. More details about the model setup and validation in the Baltic Proper is are given in (Zhurbas et al. (, 2018).

**3. 3. Results**

**3.1. OThe onset of stratification and its link to wind forcingWind forcing, hydrography and Chlorophyll *a* patterns**

To demonstrate the link between wind forcing, the onset of stratification and increase in Chl *a*, we analyzed temperature, salinity, density, and Chl *a* distributions along the gulf together with wind data infor winters 2011/12 and 2013/14. Prior to the survey of 21 December 2011, there was a s

Strong westerly wind with a , maximum along gulf wind stress was of 1.3 N m$^{-2}$, prevailed before the survey on 21 December 2011 (Fig. 34a). The Ccumulative wind stress increased by 6 N m$^{-02}$ d from 1 November to 21 December, resulting in a by 6 N m$^{-2}$ d. As a result, warm (>5 C°, Fig. 45a), relatively salty (>6.3 g kg$^{-1}$, Fig. 45b) and well-mixed water column was observed in the gulf (Fig. 45c). Very low Chl *a* concentrations were very low, , around below 1 mg m$^{-3}$, was seen in the section (Fig. 45d). Prior to the survey on 24–25 January 2012, wWeaker easterly winds had prevailed since mid January before the survey on 24–25 January 2012 (Fig. 34a). Lower temperature (3–4 C°, Fig. 45e) in the upper 20 m coincided with slightly fresher water on 24–25 January 2012 (Fig. 45f). A sSalinity minimum (down to 5.8 g kg$^{-1}$) caused stratification in the upper layer (Fig. 45g) at the a distance of 80 to 110 km in the section; this location was also characterized by , and slightly higher Chl *a* concentration (up to 1.5 mg m$^{-3}$) was seen there (Fig. 45h). Variable and relatively weak winds prevailed in late January and early February (Fig. 34a). On 7–8 February 2012, tThe temperature of the upper layer was below was lower than T$_{md}$ (2.7 °C) in the upper layer on 7–8 February (Fig. 45i), . The cold water in the upper layer coincided with lower salinity was low (4.8–6.0 g kg$^{-1}$, Fig. 45j) and there was a remamarkedable stratification and shallow UML was observed (Fig. 45k). Higher Chl *a* concentration, occasionally >2 mg m$^{-3}$, was seen in the fresher and colder water along the section (Fig. 45l). Lateral Chl *a* shape extent was closely linked to the salinity (density) structure, with h. Higher Chl *a* concentration associated with was connected to the lower salinity and vice versa.

 Westerly winds prevailed in the  period before the next survey at the end of February (Fig. 34a), resulting in  well-mixed conditions and relatively high salinity (6.0–6.7 g kg$^{-1}$) in the western part of the section on 29 February (Fig. 45m–n). Lower salinity, stronger stratification and slightly higher Chl *a* in the upper layer were observed in the central part of the section (Fig. 45n–p). The Eastern part of the section was not visited on 29 February due to ice conditions . In the middle of March (15–16 March) the water temperature was still well below T$_{md}$. and strong haline stratification was observed along the whole transect (Fig. 45r–t). Chl *a* concentrations in the upper layer were within the  range of 2–4 mg m$^{-3}$  (Fig. 45u).

Similar  trends in  wind forcing and spatiotemporal patterns  of temperature, salinity, density and Chl *a* were observed in winter 2013/14. Strong westerly wind dominated until early January 2014, with an increase in  cumulative wind stress of 10 N m$^{-2}$ d from  1 November 2013  (Fig. 34b). The 9–10 January 2014 survey shows a well-mixed water column and low Chl *a*  (56a–d). Fresher and colder water, but only slightly higher Chl *a*, Spreading ofstronger stratificationUMLwere observed inon 4–5 MarchHwas found in the cold and fresher upper layer,~~ especially in the eastern part of the section (Fig. 56i, j, l).

~~To illustrate wintertime re-stratification phenomena and formation of the shallow upper mixed layer along the gulf, we show longitudinal upper mixed layer depth, surface layer salinity, the density difference between 40 m depth and the surface layer, and surface layer Chl *a* in winters 2011/12 and 2013/2014. Very thick mixed layer (70–90 m), high surface salinity (6.3–6.5 g kg$^{-1}$), small along gulf gradient of the surface salinity, small density difference (<0.1 kg m$^{-3}$) and low Chl *a* concentration was observed on 21 December 2011 (Fig. 7). Similar characteristics were observed in most of the section on 24 January 2012. An exception was the region at a distance from 80 to 110 km, where surface salinity within the range 5.8–6.3 g kg$^{-1}$ was observed. Interestingly, saltier water was found further in the east again. Since freshwater originates from the east, the fresher water must have been first flown to west along the northern coast and later advected to the central part of the gulf likely as a filament. Upper mixed layer depth < 20 m, density difference up to 0.5 g kg$^{-1}$ and Chl *a* concentration up to 1.5 mg m$^{-3}$ were observed in this area. Mixed layer depth <20 m and density difference >0.5 g kg m$^{-3}$ occurred in most of the sections on 7–8 February, 29 February and 15–16 March 2012. Similar developments were seen in winter 2013/14. Thick mixed layer, high salinity, small density difference and low Chl *a* were observed on 9–10 January 2014. Occasionally lower salinity, smaller upper mixed layer thickness, stronger stratification and elevated Chl *a* concentration were found on 3–4 February, and well-developed stratification and Chl *a* concentration mostly in the range 2–3.5 mg m$^{-3}$ registered on 4–5 March 2014.~~

Thus,  haline stratification and elevated Chl *a* concentration was observed in both winters (2011/12 and 2013/14) from the  beginning of February. both winters (2011/12 and 2013/14). Shallow UML (<20 m) was absent after prevailing westerly winds and  when SST was >T$_{md}$. Stratification formed as fresher water occupied the upper layer.

To examine  temporal trends in  haline stratification in more detail,  we  analyzed across the gulf changes  in temperature and salinity using measurements acquired by

the Ferrybox system at along the Tallinn–Helsinki transect for in January–March 2012, 2014 and 2016 (Fig. 68). Generally, General temporal temporal changes of in salinity and temperature along the transect were in these years were quite similar for each of the study years, as was wind forcing (Fig. 34). Strong westerly winds dominated until early or mid January January, and a After the relaxation of the wind forcing, fresher water appeared was recorded in to the transect.

Based on According to the observations at the longitudinal sections (Figs. 45 and 56), the highest we assume ssea surface salinity at which stratification and relatively shallow UML can form was assumed as of 6 g kg$^{-1}$ as the highest salinity, where stratification and relatively shallow UML upper mixed layer could form. Similarly to the along-gulf observations (Fig. 45a, b), salty and warm water occupied the transect at the beginning of January in 2012 (Fig. 68a, b). The northern part of the transect was covered in fFresher water (< 6 g kg$^{-1}$) spreading from the east to west covered the northern part of the transect by the end of January, while although salinity slightly increased in the southern part of the section at this e same time. Since the main sources of freshwater are in the east, the water must have flown westward along the northern coast. The area covered by fresher water widened to almost the entire section by mid February. Water temperature declined below T$_{md}$ in the northern part in the first half of January, while in the central and southern part of the section temperature dropped below T$_{md}$ by the end of January. A similar spatiotemporal pattern in the sea surface salinity was observed in 2014 and 2016 (Fig. 68c–f). Fresher water first appeared in the northern part in the first half of January in early or mid January both in 2014 and 2016. The onset of haline stratification took a place occurred slightly earlier in 2016 due to . This is associated with the wind forcing – the westerlies had eased off already by at the end of December 2015 (Fig. 3c). The segment covered by fresher water widened during January and most of the transect was occupied by water with salinity <6 g kg$^{-1}$ at the end of January 2016 and in the mid February 2014. A pulse of sStrong westerly wind occurred impulse occurred at the end of January–beginning of February in 2016 (Fig. 34c). We suggest that the lighter, less saline water that originates in the from the east flowed westwards aloalong the northern coast to the west and was later transported toward the southern coast in the central and western part of the gulf. The lLatter is likely related to the the Ekman transport induced by the westerly winds wind impulse early February in both years (Fig. 34). Thus, stratification related to the spreading of fresher water forms about one month earlier in the northern part of the gulf than in the than in the southern part of the gulf.

**3.2. SThe occurrence spatiotemporal patterns of the restratification stratification phenomenona**

Here, Next, we describeexamine The the spatiotemporal pattern of the restratification stratification process using can be described by model simulation data and statistics of statistics of historical observations. Monthly mean simulated UML depth and occurrence of the UML depth <20 m are presented in Fig. 9.

As noted from the in situ observations, the haline stratification forms after the relaxation of the westerly winds. The annual cycle of the along-gulf component of wind stress shows higher monthly mean values (>0.04 N m$^{-2}$) and higher variability from October–January (Fig. 7); this . It means that the strong westerly winds from westerly directions are more frequent and storminess is higher in these months. GenerallyAs a consequence, the occurrence of UML depth <20 m was infrequent and as very low and mean UML depth varied between within

the range 40–60 m in the western and central gulf in November, December and January 2010–2019 (Fig. 89a f). As an exception, the probability of the UML depth <20 m was 30–40% in the northern part of the eastern area in January. Winds from the west are weaker and storms are less frequent in February and March (Fig. 8). In February, the occurrence occurrence of UML depth <20 m increased to 50–60% (Fig. 98g and h), although . Still, in the southern and western parts of the gulf, the mean UML depth was 30–40 m in February. These statistics from based on model simulation data well agree well with our observations of westward advection of fresher water from the northern coast (Fig. 68). The Mmean UML depth was 20 m or lower in the central part of the gulf in March, and while the UML was thicker at the gulf entrance (Fig. 8) area of the gulf in March ; (Fig. 89i). Tthe occurrence of the UML depth <20 m was >60% in the central part, around 50% at the gulf entrance area of the gulf and much lower to the ess in the west of from the longitude 22° E (Fig. 89j). A similar pattern is shown in 
[revised manuscript text omitted]
, All these studies were concerned with stratification dealt with the phenomenaon under the ice, whereas in our . In the current study , on the example of the Gulf of Finland, we have shown ed that wintertime stratification may could also occur in at the basin-scale (along-gulf extent 400 km) and in the absence of without considerable ice coverage. During the onset of restratification-stratification in late January 2012, the Gulf of Finland was virtually ice-free. In both winters (2011/12 and 2013/14) at the end of January, only the eastern part of the Gulf of Finland and the adjacent part of the northern shore of the gulf were ice covered with iceat the end of January; t. Thus, winter stratification phenomenon occurred even ifwhen most of the Gulf of Finland was not covered by ice. It should be has to be noted that, besidesalong with the frequency of easterly winds, also ice coverage is larger in the case of low NAO index is also associated with increased ice coverage (Jaagus, 2006). The landfast ice zone would be is expected to prevent vertical mixing and therefore supports lateral advection of riverine fresher water (Granskog et al., 2005).

The western border of the stratification phenomenon is around 23° E, i.e. at the entrance to the gulfentrance area to the gulf between Hiiumaa Island and the Finnish coast. This . It means Vertical vertical mixing dominates over the lateral buoyancy fluxes in the Baltic Proper, and shallow stratification is not a common feature. in the Baltic Proper. The Tabshe absenccence of the phenomenon in the Baltic Proper can be explained by the long distance from the rivers, due to its larger size and and due to the ttopography. TheR riverine input per unit area in the Gulf of Finland is 7–8 times larger than in the Baltic Proper (Leppäranta and Myrberg, 2009). As the One can note that the wintertime stratification phenomenon vanishes at the wider entrance area to in the area, where the extension of the Gulf of Finland, it is . gets wider. Thus, likely that the 
[revised manuscript text omitted]

800

---

## Referee Report (RR1)

**2nd review of "Winter stratification phenomena and its consequences in the Gulf of Finland, Baltic Sea" manuscript OS-2020-40**

**General comments**

The authors did a decent job in addressing my comments, and I am glad to see that the manuscript improved since my first read. However, I still have some concern regarding the quality of the writing and the figures. Some parts of the text, particularly the paragraphs introduced after the revision, contain some repetitive sentences and feel a bit jumbled. I also have a pseudo-major comment that I missed in the first round. I have the feeling that one of the main points of the manuscript is that wintertime haline stratification resulted in enhanced chlorophyll-$a$ concentrations. You show this very nicely with the cruise data. Then you make a thorough effort to assess the generality and the spatial coverage of the haline stratification with different datasets, but there is no equivalent assessment for phytoplankton bloom or chlorophyll dynamics. The biological implications of your results is thus much weaker than the description of the physical rationale. Would it be possible to use a satellite product to expand the spatio-temporal coverage of chlorophyll observations? Below, I list a series of specific comments concerning mainly writing and presentation issues, but I may be missing a significant number of them. I suggest a careful inspection of these aspects before publication.

**Specific comments**

**Abstract.** Please revise the abstract language. It feels patchy and contains repetitive sentences. For example the "relaxation of westerly winds" is mentioned twice to give a very similar message.

**Line 12** "In this study, we demonstrate that wintertime UML stratification is common in the Gulf of Finland.". This sentence is contradictory, if it is a mixed layer is not stratified by definition. I recommend formulating it differently.

**Line 15** "Fresher water and haline stratification occurs". That sounds a bit weird to me. Reformulate.

**Line 154** Add space after "simulations".

**Line 206** "m$^{-02}$" $\rightarrow$ "m$^{-2}$"

**Lines 256-258** I think this sentence about the mechanism driving the spreading of haline stratification belongs more to the discussion. You need to support it with references. There is another interesting point that emerges from Figure 11b. There are a number of eddy-like frontal instabilities that seem spread shallow mixed layers towards the south. It may be worth mentioning this in the discussion. See for example: `https://science.sciencemag.org/content/337/6090/54.full?rss=1`

**Line 287** "Restratification phenomenon were". Phenomenon is singular, so "was".

**Lines 291-305** This part is a bit dense and hard to follow sometimes. Consider rephrasing a bit.

**Section numbering** Section numbering is wrong for Discussion and Conclusions

**Line 327** "Deepen the mixed layer depth". I would remove "depth".

**Lines 341-357** This paragraph is also a bit jumbled and contains typos like "accounts contributes". In line 344, it is not clear to me what you mean by "topography".

**Line 359** The meaning of "occasionally" here is blurry. Please try to be more precise.

**Lines 390-393** This sentence is weirdly constructed.

**Lines 395** "We can assume" doesn't sound very convincing to me, maybe "Therefore, our results suggest/indicate"

**Figure 4, caption** There is a double ".." after 2911/12. Also 2911 is wrong.

**Figure 6** The tick labels of the colorbar overlap with the ticklabels on the y-axis.

**Figure 10a** It is extremely difficult to extract information from this new panel, too many superimposed lines. Please consider improving this.

**Panel labelling** The labelling of the different panels (a, b, c, etc) is located in different places for the different figures, sometimes it is inside the figure, sometimes in the panel title, others in the y-axis label. It would be better for the reader to homogenise this.

---

## Author Response (AR2)

Editor comments

**Comment:** Please carefully check the reference to the figures. Figure 2 is not referred to in the text. Probably where Figure 3 is referred to for the first time, this should be Figure 2. Also other references to figures may be incorrect. Please check.

**Reply and action:** We have checked carefully the reference to figures and made corrections where necessary.

**Comment:** Please be certain to have all geographic names used in the text are also shown on the map of figure

**Reply and action:** We added all the names mentioned in the text to figure 1.

**Comment:** L35-36 „determine vertical fluxes between the surface and sub-thermocline layer." Please add what kind of vertical fluxes. Of water, particles, organic matter, etc.?

**Reply:** We removed the previous sentence and added, what we mean.

**Action:** It reads now "Characteristics of the seasonal pycnocline (e.g. strength) determine vertical physical (e.g. heat and salt), biogeochemical (e.g. oxygen, nutrients), biological (e.g. plankton) or pollution (e.g. microplastics) fluxes between the surface and sub-thermocline layer."

**Comment:** L36-37 "Moreover, the vertical structure of currents is strongly linked to pycnoclines (Suhhova et al., 2018)." This is not clear to me. Please expand and/or explain.

**Reply:** We made the sentence more specific.

**Action:** "Moreover, the current shear maximum is strongly linked to the vertical location of a seasonal pycnocline (Suhhova et al., 2018)."

**Comment:** L46 "approximately" instead of "virtually"

**Reply and action:** done.

**Comment:** L49-50 Please rearrange the references. Double parentheses are not necessary here.

**Reply and action:** done.

**Comment:** L133 Please define OSTIA

**Reply and action:** done.

**Comment:** L136 Insert year for the reference

**Reply and action:** Since we have added the most up to date reference related to OSTIA (Good et al 2020) during revision, this sentence and reference are not needed anymore in the manuscript.

**Comment:** L206 m-2 and delete: d

**Reply and action:** We fixed m-2, but d (day) should stay there. The unit is N m$^{-2}$ d.

**Comment:** L303-304 "Thus, large scale atmospheric forcing alters the restratification." This is not correct, I think. The large-scale forcing provides conditions that are beneficial for restratification.

**Reply:** Yes that is a good point, it was not well expressed.

**Action:** We changed to "Thus, large scale atmospheric forcing provides conditions for the restratification process."

**Comment:** L307 4. Discussion (correct numbering).

**Reply and action:** done.

**Comment:** L330 phenomena (as the sentence is in plural)

**Reply:** We talk about process here (singular). According to our understanding, it should stay the same, as in the title.

**Action:** No action.

**Comment:** L336-337 "thus, winter stratification phenomenon occurred even when most of the Gulf of Finland was not covered by ice." Exactly this has been written some lines above, so can be deleted here.

**Reply and action:** done.

**Comment:** L342 Please refer to Figure 1 here

**Reply and action:** done.

**Comment:** L347 I think "accounts" should be deleted here.

**Reply and action:** done.

**Comment:** L384 5. Conclusions (change numbering)

**Reply and action:** done.

**Comment:** L425 I think this means anonymous, right? Please use that term.

**Reply and action:** Yes and we changed it to anonymous.

**Comment:** L432-434 This reference is incomplete (no journal etc.), and with double terms. http and doi is the same. Please correct

**Reply and action:** done.

**Comment:** L436-438 No journal etc., and double link

**Reply and action:** done.

**Comment:** L447 Oceans not: Ocean.

**Reply and action:** done.

**Comment:** L448 Oceans not: Ocean

**Reply and action:** done.

**Comment:** L455-456 What kind of publication is this? This link is not valid. Please give more info and correct.

**Replay:** It is Fennel, K.: Convection and the timing of phytoplankton spring blooms in the western Baltic sea, Estuar. Coast. Shelf Sci., 49(1), 113–128, doi:10.1006/ecss.1999.0487, 1999.

**Action:** We fixed the text.

**Comment:** L461 Hydrobiologia, 554, 57–65, 2006. (change format)

**Reply and action:** Done.

**Comment:** L484 iden dito, change format

**Reply and action:** Done.

**Comment:** L488-490 Please provide more info on this publication. The link does not lead to this paper.

**Reply and action:** Done.

**Comment:** L516-517 Provide editor of book

**Reply:** they are both authors of the chapter and the editors of the book.

**Action:** We provided.

**Comment:** L528 Page numbers missing

**Reply and action:** we added.

**Comment:** L573-574 Please provide more info on this reference

**Reply and action:** we provided.

**Comment:** L603-605 Please provide more info on this reference, delete: n.d.

**Reply:** since we have a provided a more up to date reference (Good et al. 2020) during revision, this reference is not needed anymore.

**Action:** we removed this reference.

**Comment:** L618 Change date format 1 November etc.

**Reply and action:** done.

**Comment:** L627 caption Figure 4: These are not profiles but sections. What is called profile should be a transect.

**Reply and action:** We fixed.

**Comment:** Figure 4: It is strange to have the color scale in one of the panels. Please place it outside.

**Reply and action:** We fixed.

**Comment:** Figure 5 idem ditto

**Reply and action:** We fixed, except temperature. Because the color scale is not same in all panels.

**Reviewer comments**

General comments

**Comment:** The authors did a decent job in addressing my comments, and I am glad to see that the manuscript improved since my first read. However, I still have some concern regarding the quality of the writing and the figures. Some parts of the text, particularly the paragraphs introduced after the revision, contain some repetitive sentences and feel a bit jumbled. I also have a pseudo-major comment that in missed in the first round. I have the feeling that one of the main points of the manuscript is that wintertime haline stratification resulted in enhanced chlorophyll-a concentrations. You show this very nicely with the cruise data. Then you make a thorough effort to assess the generality and the spatial coverage of the haline stratifcation with different datasets, but there is no equivalent assessment for phytoplankton bloom or chlorophyll dynamics. The biological implications of your results is thus much weaker than the description of the physical rationale. Would it be possible to use a satellite product to expand the spatio-temporal coverage of chlorophyll observations? Below, I list a series of specific comments concerning mainly writing and presentation issues, but I may be missing a significant number of them. I suggest a careful inspection of these aspects before publication.

**Reply:** Thank you for your time and great suggestions again!

**Action:** We have addressed all the specific comments below. However, we are not able to perform an equivalent assessment for the chlorophyll-a dynamics. The main reason is unreliable remotely sensed Chl *a* data in this area. The standard products that are available in the CMEMS for instance are not accurate enough for the gulf. The Gulf of Finland is a special area and needs dedicated algorithms. Regional algorithms for spring and summer conditions have been developed here (e.g. https://www.sciencedirect.com/science/article/pii/S0078323414500449?via%3Dihub). To get specific algorithms for winter conditions is too big work to include in the present paper. Likely this work needs a separate paper itself. We think it is a good idea to include the satellite observations and we considered it already before submission, but we prefer to leave this work for the next publications.

Specific comments

**Comment:** Abstract. Please revise the abstract language. It feels patchy and contains repetitive sentences. For example the \relaxation of westerly winds" is mentioned twice to give a very similar message.

**Reply:** Yes, we agree.

**Action:** We revised the abstract according to your critics.

**Comment:** Line 12 \In this study, we demonstrate that wintertime UML stratifcation is common in the Gulf of Finland.". This sentence is contradictory, if it is a mixed layer is not stratifed by defnition. I recommend formulating it differently.

**Reply:** Yes, we agree.

**Action:** It reads now: „In this study, we demonstrate that wintertime shallow stratification is common in the Gulf of Finland."

**Comment:** Line 15 \Fresher water and haline stratifcation occurs". That sounds a bit weird to me. Reformulate.

**Reply:** Yes, we agree.

**Action:** It reads now: „Haline stratification emerges"

**Comment:** Line 154 Add space after \simulations".

**Reply and action:** fixed

**Comment:** Line 206 \m$^{-02}$" to \m$^{-2}$"

**Reply and action:** fixed.

**Comment:** Lines 256-258 I think this sentence about the mechanism driving the spreading of haline stratification belongs more to the discussion. You need to support it with references. There is another interesting point that emerges from Figure 11b. There are a number of eddy-like frontal instabilities that seem spread shallow mixed layers towards the south. It may be worth mentioning this in the discussion. See for example: https://science.sciencemag.org/content/337/6090/54.full?rss=1

**Reply:** Yes, we agree, it rather belongs to the discussion. We think it is very important point you have made about eddies. Thank you"

**Action:** We removed that part from the results and merged it with the discussion. We mention the eddy-like features now in discussion. Moreover we added another panel (middle panel b) to the fig to discuss a bit more on that topic. The panel a shows the situation, when westerly winds dominate (as in previous version of the manuscript). Panel b (new) shows the situation after the dominance of easterly winds: advection along the northern coast. Panel c shows the development after short westerly wind

impulse. Thus, panel c explains how shallow mixed layer spreads to south, the precondition for the situation c is the „b", i.e. fresher water along the northern coast. This temporal development well agrees with our Ferrybox observations (Fig. 6): fresher water first appears to the northern coast and after westerly wind impulse it spreads to the south.

**Comment:** Line 287 \Restratifcation phenomenon were". Phenomenon is singular, so \was".

**Reply and action:** fixed.

**Comment:** Lines 291-305 This part is a bit dense and hard to follow sometimes. Consider rephrasing a bit.

**Reply and action:** We rephreased some sentences here and hope it is better now.

**Comment:** Section numbering is wrong for Discussion and Conclusions

**Reply and action:** fixed.

**Comment:** Line 327 \Deepen the mixed layer depth". I would remove \depth".

**Reply and action:** fixed.

**Comment:** Lines 341-357 This paragraph is also a bit jumbled and contains typos like \accounts contributes". In line 344, it is not clear to me what you mean by \topography".

**Reply:** Yes, we agree.

**Action:** We have fixed the error and specified what we meant by tropography. We also made some other small changes to make it better to read.

**Comment:** Line 359 The meaning of \occasionally" here is blurry. Please try to be more precise.

**Reply and action:** We agree and found the word „occasionally" indeed is not necessary here.

**Comment:** Lines 390-393 This sentence is weirdly constructed.
**Reply and action:** We changed it. We believe it is now better to read.

**Comment:** Lines 395 \We can assume" doesn't sound very convincing to me, maybe \Therefore, our results suggest/indicate"
**Reply:** We agree
**Action:** We changed as you suggested.

**Comment:** Figure 4, caption There is a double \.." after 2911/12. Also 2911 is wrong.
**Reply and action:** We fixed.

**Comment:** Figure 6 The tick labels of the colorbar overlap with the ticklabels on the y-axis.
**Reply:** We fixed.

**Comment:** Figure 10a It is extremely difficult to extract information from this new panel, too many super-imposed lines. Please consider improving this.
**Reply:** We made it easier for a reader.
**Action:** We kept only two highest NAO years and two lowest NAO years to illustrate the impact on the time-series of the UML in the Gulf of Finland. We also mention the selected years in the text (end of the results chapter). We kept all years for the Gotland Deep, but the same color.

**Comment:** Panel labelling. The labelling of the different panels (a, b, c, etc) is located in different places for the different figures, sometimes it is inside the figure, sometimes in the panel title, others in the y-axis label. It would be better for the reader to homogenise this.

**Reply and action:** We homogenized.

[revised manuscript text omitted]